# High Concentrations of Ice Crystals in Upper Tropospheric Tropical Clouds: Is there a Link to Biomass and Fossil Fuel Combustion?

Graciela B. Raga[1], Darrel Baumgardner[2], Blanca Rios[1], Yanet Díaz-Esteban[1], Alejandro Jaramillo[1], Martin Gallagher[3], Bastien Sauvage[5], Pawel Wolff[5] and Gary Lloyd[3,4]

[1]Centro de Ciencias de la Atmósfera, Universidad Nacional Autónoma de México, Mexico City, Mexico
[2]Droplet Measurement Technologies, LLC, Longmont, CO, USA
[3]Centre for Atmospheric Science, University of Manchester, Manchester M13 9PL, UK
[4]National Centre for Atmospheric Science (NCAS), University of Manchester, Manchester M13 9PL, UK
[5]Laboratoire d'aérologie (LA), CNRS UMR-5560 et Observatoire Midi-Pyrénées, Université de Toulouse, France

*Correspondence to: Darrel Baumgardner (Darrel.baumgardner@gmail.com)*

**Abstract.** Eight years of upper tropospheric (UT) ice crystal measurements with the Backscatter Cloud Probe (BCP), installed on commercial aircraft operated as part of the In-Service Aircraft for a Global Observing System (IAGOS), have been analyzed to assess the frequency and characteristics of extreme ice crystal events (EIE), defined in this study as encounters with clouds that have number concentrations exceeding 5000 L$^{-1}$. A total of 3196 events, in clouds of horizontal extent $\geq$ 2.5 km, were identified during the period from December 2011 to March 2020 in the latitude band between 30°S and 30°N. Regions of anthropogenic sources of carbon monoxide, with particles that can alter cloud microphysics, were attributed to these EIE in UT clouds using the SOFT-IO model. The evaluation of low- and upper-level kinematic variables from the European Centre for Medium-Range Weather Forecasts (ERA5) reanalysis, combined with spatial distributions of aerosol optical depth and regions of biomass burning, highlight the physical mechanisms by which the particles are lofted to flight levels in regions of deep convection. The maps of lightning frequency, derived from the World Wide Lightning Location Network (WWLLN), provide additional evidence of the role of deep convection in transporting aerosol particles, cloud hydrometeors and carbon monoxide to aircraft cruising altitudes. The evaluation of aerosol particle mass concentrations and composition from the Modern-Era Retrospective analysis for Research and Applications, Version 2 (MERRA-2) contributes additional evidence for a link between regions of EIE and surface emissions of dust, black carbon (BC), organic carbon (OC) and sulfate particles. Given the composition of the source aerosols and the role of deep convection in their transport to the UT, the sampled ice clouds likely originate from the homogeneous or heterogeneous freezing of droplets formed on these particles, as have been reported in previous studies. The results from this study, which have been obtained from a large sample of measurements, have ramifications related to satellite measurement validation, weather forecasting and climate change. In addition, over 2000 of the randomly sampled clouds had derived ice water contents larger than 1 gm$^{-3}$, a concentration that is considered potentially hazardous to commercial aircraft operations.

Keywords: Ice crystal concentrations, IAGOS, biomass burning, urban pollution, deep convection, ice nucleating particles, immersion freezing, high ice water content, carbon monoxide

**1.0      Introduction**

The studies reported by Krämer et al (2016, 2020) provide a comprehensive description of the origins and microphysical properties of cirrus clouds based on 150 flights with nine research aircraft that accumulated 168 hours of cloud measurements. These data were acquired during multiple field campaigns in the mid-latitudes and tropics

over a period of 19 years (1999-2017). Statistics were gathered on the number concentration ($N_{ice}$), ice water content (IWC) and mass mean radius ($R_{ice}$) as a function of ambient temperature and compared to 10 years of cirrus ice number concentrations derived from satellite measurements. This extensive data set serves to underscore the importance of aircraft measurements, not only to document cirrus properties, but also to provide the detailed information about those cloud characteristics that cannot be extracted from the lower spatial resolution satellite data.

Furthermore, *in situ* measurements have provided the critical information needed to refine parametrizations and improve how cirrus clouds and their microphysical and optical properties are represented in climate models (Kärcher, 2018; Kärcher et al., 2010).

As invaluable as targeted field campaigns are for addressing specific scientific questions related to cloud formation and evolution, as noted by Beswick et al. (2015) the results from such programs are by necessity limited to relatively

short time periods and small cloud regions. In addition, while providing essential information on the microphysical properties, the sampling is usually done through the region of the cloud that is visually most developed, i.e., away from cloud edges. This implies that the results from the measurements are not necessarily the most indicative of the more general population of similar cloud types. Hence, a much wider ranging data set is needed, temporally and geographically, to provide corroborative data to extend the data sets beyond limited research field campaigns.


The European Research Infrastructure program IAGOS (In-service Aircraft for a Global Observing System) was developed with the primary objective of building a database of global measurements, over an extended period of time, as a source of information that would be truly representative across all seasonal and spatial scales (Petzold et al., 2015). Although the instrumentation on the IAGOS aircraft fleet is not as comprehensive as that found on the

typical cloud research aircraft (Petzold et al., 2015), Beswick et al. (2015) demonstrated that the cirrus cloud microphysical statistics, acquired with the cloud probe installed on the IAGOS aircraft, were in good agreement with those that were summarized in published studies compiled from aircraft-based research campaigns. This leads us to conclude that data taken during the IAGOS flights should be considered as an integral and complementary component of the global data set of cirrus measurements, as summarized by Krämer et al (2016, 2020).


As discussed by Krämer et al. (2016) the ice crystals in upper tropospheric, lower stratospheric clouds (UTLS) can form by two mechanisms: directly by depositional ice nucleation (*in situ* origin) or carried in updrafts from lower altitude, mixed phase clouds via freezing of liquid droplets (liquid origin), either homogeneously or heterogeneously. From their modeling of ice clouds and analysis of measurements made in many parts of the world

they conclude that the clouds with the highest ice crystal concentrations are of liquid origin, found most often at tropical latitudes. In their analysis of three years of IAGOS measurements Beswick et al. (2015) identified frequent events in which $N_{ice}$ exceeded 10000 $L^{-1}$, a concentration that is > 300 times the median values and 30 times larger than the 90[th] percentile reported by Krämer et al. (2020) for $N_{ice}$ measured *in situ* and derived from CALIPSO and CloudSat measurements (Sourdeval et al., 2018). Beswick et al, (2015) also observed that more than 60% of the

high ice concentration clouds encountered were found over the tropical Atlantic, Southeast Asia or the region centered in Papua/New Guinea. Hence, based on the studies by Krämer et al. (2016), it is reasonable to conclude that the high ice crystal concentrations reported by Beswick et al. (2015) in tropical regions were most likely formed via the freezing of droplets on ascent to the UTLS. Note that homogeneous and heterogeneous freezing are not mutually exclusive, i.e., supercooled droplets lofted by vigorous updraft may initially freeze heterogeneously at temperatures

warmer than -38° and then those that remain unfrozen will freeze homogeneously as they cool below that that temperature.

      Depositional ice nucleation will also lead to ice crystal formation even when droplet freezing may be the dominant mechanism Many laboratory and field studies have measured the concentration of ice crystals, formed from INP

composed of a wide range of materials, as a function of temperature. The average trend is that the number of activated INP increase as temperature decreases (Kanji et al., 2017). The highest concentrations that have been observed by this nucleation pathway is around 1000 $L^{-1}$, or about a factor of 10 fewer than the highest concentrations of ice crystals measured by Beswick et al. (2015).

Secondary ice particle (SIP) formation has also been invoked as a process that can produce concentrations much greater than predicted from the INP versus temperature parameterizations (Field et al., 2017). The majority of the proposed mechanisms occur at much warmer temperatures than those in the UTLS; however, if secondary ice crystal production is active within strong updrafts, they could be a possible source of high ice crystal concentrations at the cruising altitudes of the IAGOS aircraft.


      Regardless of how they are formed, two essential components that are required for liquid origin clouds are: 1) aerosol particles that can activate as cloud droplets (CCN) or ice crystals (INP) and 2) strong vertical motions to carry the ice to the UTLS. Vigorous updrafts also are accompanied by supersaturations that can be large enough to activate as cloud droplets even very small and less hygroscopic aerosol particles (Fan et al., 2018) . Hence, deep

convection, given its ubiquitous presence in the topics, is the most likely mechanism for lifting ice of liquid origin to the IAGOS flight levels. We may then ask: What are the most common sources of the copious aerosol particles resulting in cloud droplets and ice crystals in tropical regions?

      Beswick et al. (2015) suggested that the high ice crystal concentrations detected in the maritime-continental, tropical

region over Malaysia and Indonesia were possibly linked to biomass burning. Observational and theoretical studies have shown how gaseous and particulate emissions, from not only biomass burning (BB), but also from urban

pollution (UP), can reach aircraft cruising altitudes where ice crystal clouds are encountered (Osman et al., 2016; Ditas et al., 2018; Hirsch and Koren, 2021). Although some fraction of the particles emitted from biomass and fossil fuel combustion will act as CCN or INP (Barry et al., 2021; Brooks et al., 2014; Gorbunov et al., 2001; Grawe et al.,

2016; Knopf et al., 2018; Jahl et al., 2021; Popovicheva et al., 2008; Schill et al., 2016; Umo et al. 2015) as a result of their chemical composition, as concluded by Fan et al. (2018), when updrafts are sufficiently strong, the resulting supersaturations can activate the small combustion particles as water droplet

Anthropogenic emissions from fossil and biofuel combustion include both aerosols and carbon monoxide. Ding et

al. (2015) used spaceborne Measurements of Pollution in the Troposphere (MOPITT) data, in combination with aircraft measurements from the Measurement of Ozone and Water Vapor by Airbus In-Service Aircraft (MOZAIC) program to document the presence of anomalously high carbon monoxide (CO) in the upper troposphere over East Asia. They used back-trajectory analysis to link anthropogenic sources over the Sichuan Basin and the North China Plain as well as from forest fires over Indochina. The CO was transported through strong frontal lifting, embedded

deep convection and by orographic forcing.

Osman et al. (2016) developed a climatology of CO using trajectory mapping of data from MOZAIC and IAGOS that linked biomass burning in central and southern Africa and anthropogenic emissions in eastern China to upper tropospheric CO anomalies. Similar results are reported by Sheel et al (2016) over central India.


Yamasoe et al. (2015) employed the MOZAIC/IAGOS database, complemented with model simulations, to link the contributions of CO in the UTLS to local, Central and North American, Caribbean and African anthropogenic and BB emissions. Ditas et al. (2018) encountered frequent BB plumes in the UTLS over the Atlantic Ocean. The refractory black carbon (rBC) mass concentration in these plumes was measured by the Single Particle Soot

Photometer (SP2; Baumgardner et al. 2005). The average rBC concentrations in these plumes were more than 20 times higher than background values. Nearly all of the rBC particles sampled in the UTLS were covered with a thick coating, an indicator of aging that increases the CCN activity of BC; however, even without a hygroscopic coating this BC particles can be activated is the supersaturation is sufficiently high.

The importance of the Asian Summer Monsoon (ASM) for lofting anthropogenic pollution and forming the Asian Tropopause Aerosol Layer (ATAL) is discussed by Vernier et al. (2015). Aircraft measurements identified particles composed of carbonaceous and sulfate materials and CALIPSO observations indicated that deep convection associated with the ASM transports surface pollution particles to the ATAL.

Although combustion does produce both CO and aerosol particles, the processes that remove CO are different than those that remove aerosols; hence, CO concentrations cannot be used as a direct proxy for particle concentrations. The CO can, however, be used as a tracer of air masses that are transporting both CO and particles.

The goal of the study reported here is to improve our understanding of the processes that lead to the formation of high ice concentration clouds in the UTLS using an analysis of the IAGOS database of measurements over the period December 2011 to March 2020. A preliminary evaluation of this data set showed that ice number concentrations ($N_{ice}$), larger than 5000 L$^{-1}$, fall in the 85$^{th}$ percentile of all upper tropospheric (UT) ice clouds encountered, a value that is more than 10 times higher than the 90$^{th}$ percentile reported by Krämer et al. (2020) of 300 L$^{-1}$. Note that while the previously mentioned in situ studies refer to clouds in the UT as cirrus, we wish to clarify that some percentage of the clouds that we report in this study are likely measured in the upper regions of cumulonimbus and in their outflow anvils. Hence, from this point onward, we will refer to all the clouds measured in our study as "ice clouds" and to all cloud encounters when ice crystal concentrations exceed 5000 L$^{-1}$ as *Extreme Ice Events* (EIE). This latter definition of an EIE is created with reference to the median and 90$^{th}$ percentile values reported from previous in situ and satellite studies. The term EIE should not be confused with or equated to the term *Ice Crystal Icing* (ICI), a definition used by the community of researchers who study how ice crystals impact aircraft performance. ICI is usually associated with ice water content (IWC) and not with $N_{ice}$, although the two variables are obviously linked. The present study focuses on understanding the processes that result in high $N_{ice}$ and on the identification of the possible sources of the precursor aerosols that form these crystals. We have specifically chosen to use $N_{ice}$ rather than IWC as our analysis metric because of the large uncertainties (> ±50%) associated with deriving IWC from the size distributions measured with the BCP compared with the estimated ±15% uncertainty in $N_{ice}$ (Beswick et al., 2014). We do, nevertheless, calculate IWC and discuss it within the context of ICI and aircraft operations at the end of the discussion section.

Figure 1 shows the locations of EIE observed within the ± 30$^0$ zonal band from December 2011 through March 2020, along with megacities, i.e., those urban areas with populations larger than 10 million. All worldwide flight tracks from IAGOS aircraft up to 4 October 2021 are shown in Fig. SM1, in the supplementary material. The EIE are clustered along the flight routes from Europe to South America and to Africa, but the majority of events are found in the large cluster centered around Taiwan and radiating towards India, Japan, Indonesia and Australia/Oceania. The color-coded markers indicate the ice crystal number concentration of the EIE and, while the majority of concentrations range between 5000 and 20000 L$^{-1}$ (purple markers), there are a number of regions where concentrations exceeding 50000 L$^{-1}$ were detected, as indicated by green, yellow and red circles. The colored, dashed lines demarcate the four regions into which the events are grouped in our analysis and that encompass different regions of BB and UP.

This study intends to show that the high ice concentrations encountered along commercial flight tracks are frequently within regions where there is also evidence of emissions from BB and UP as sources for aerosol particles that intervene in cloud microphysical processes. The approach that we are taking is to gather complementary data from multiple sources to build the case in support of our conjecture that BB and UP are the most likely sources, not only for the EIE, but for most of the ice clouds encountered at these latitudes. An analysis of *in situ* cloud and CO measurements, coupled with back-trajectory modeling of the air masses that transport CO, identifies the possible

sources of CCN for water droplets and ice crystal nuclei. The complementary analysis compares the location of the EIE with the climatological patterns of upper-level divergence (an indicator of regions with upward vertical motion), lightning, outgoing long-wave radiation (OLR, indicative of the presence of clouds), aerosol optical depth (AOD) and regional composition of aerosol particles in the UT.


The four objectives of the study are to: 1) document the frequency of EIE by geographic region within the tropical latitude band most impacted by BB and UP emissions, 2) evaluate the seasonal variations of EIE as related to dry and rainy periods, 3) identify regional sources of CO and aerosols associated with the EIE and 4) show that there is sufficient convection to transport aerosol and the cloud particles that form on them, up to cruising altitude.


## 2.0     Measurement and Analysis Methodology

Measurements of CO mixing ratios and ice crystal concentrations were made with sensors installed on eight Airbus
A330/A340 IAGOS aircraft. The data analyzed cover the period from December 2011 to March 2020, a total of 22857 flights, during which 67172 ice clouds were sampled, accumulating 1483 hours in clouds. The ice crystal measurements are made with Backscatter Cloud Probes (BCP; Droplet Measurement Technologies, LLC), whose detection principles are described by Beswick et al (2014). The BCP is mounted inside the aircraft and interfaces to a window plate that also holds the inlet that brings ambient air to the gas analyzers (Nédélec et al., 2003;2016). The
BCP transmits a focused, collimated laser beam through a window and ice crystals that intersect this beam then scatter light. Some fraction of the backscattered light is collected through another window and the collected photons are converted to an electrical signal that is processed and used to derive an equivalent optical diameter (EOD). The nominal size range is 5 – 90 µm with an estimated uncertainty of ±30% when deriving the EOD of ice crystals (Bestwick et al., 2014). Detected crystals are binned by size into a 10-channel frequency histogram that is
transmitted to the data acquisition system every four seconds.  The number concentration is calculated as the number of particles detected divided by the volume of sampled air. The estimated concentration uncertainty is approximately ±15% (Bestwick et al., 2014). The IWC is also derived from the size distributions but the uncertainties in the shape and density of ice crystals leads to an estimated uncertainty of ±50%. The BCP samples 250 $cm^3$ every four seconds, a spatial resolution of approximately one kilometer at normal cruise altitude airspeeds of 250 $ms^{-1}$ for Airbus
A330/A340 IAGOS aircraft.

The ice crystal concentrations reported in this study are the maximum values from individual clouds, where clouds are identified as those encountered at an altitude $\geq$ 8000 m and that have a measured $N_{ice} \geq$ 50 $L^{-1}$. This latter threshold is imposed to ensure that the number of crystals sampled are statistically representative of a cloud's parent
population with an uncertainty < 30% (Beswick et al., 2015; Lloyd et al., 2020).

Carbon monoxide (CO) mixing ratios are measured with a Thermo Scientific Model 48 that has been modified to improve its performance, including the use of an optimized IR detector, cell pressurization, air drying and frequent zeroing (Nédélec et al.,2003). A 30-s integration time is used to obtain a minimum detection limit of 10 ppbv and thus, the CO samples are obtained at a horizontal resolution of about 7.5 km at cruise speeds.


The sampled air mass histories are evaluated using the FLEXPART Lagrangian particle dispersion model (Stohl et al., 2005) that simulates the long-range and mesoscale transport of tracers released from point, line, area or volume sources. The CO mixing ratios detected at cruising altitude are linked to emissions of fossil fuel combustion from urban regions and from biomass burning using the SOFT-IO model, as described in detail by Sauvage et al. (2017a). The SOFT-IO model utilizes the Emissions of Atmospheric Compounds & Compilation of Ancillary Data (ECCAD) database that documents inventories of global emissions from UP (MACCity) and BB (GFAS). Calculating the contributions of CO emissions for each point along the flight tracks requires three kinds of data: the residence time from backward transport calculated with FLEXPART, the CO surface emissions from the ECCAD database and the injection profile that defines the fraction of pollutants diluted at each vertical levels  just after emissions. The interested reader is encouraged to find more detail of this very useful model in Sauvage et al. (2017a).



For every IAGOS flight from 2011 to 2020, when the CO sensor was operational, the FLEXPART back trajectories were computed at each sampling point along the flight paths, estimating the location of the sampled air as far back as 20 days prior to its arrival at the aircraft. The model incorporates the wind fields from the 6-hourly operational analyses and 3-hour forecasts by the European Centre for Medium-Range Weather Forecast (ECMWF).


We are particularly interested in identifying anomalous CO mixing ratios within the database that would indicate plumes originating from surface UP or BB emissions. Mixing ratios greater than the climatological background along the aircraft tracks are defined here as "anomalous" and Sauvage et al. (2017a) identify these anomalies through a methodology implemented to detect pollution layers in the IAGOS database (Petetin et al., 2018; Cussac et al., 2020). One of the products that is available through the IAGOS central database is flight-by-flight netCDF files that provide the CO mixing ratio that has been transported from either biomass burning or urban regions and that exceeds the background (Sauvage et al., 2017a, b). For the UT the background CO mixing ratio is determined by calculating seasonal median values, over the entire IAGOS database, for 10 different regions worldwide. These background CO values are subtracted from each SOFT-IO simulated CO mixing ratio every four seconds along the flight track; the anomaly $\Delta CO > 0$, in units of ppbv, are indicative of the possible influence of surface emissions of UP and/or BB.



The data files with the SOFT-IO simulations are archived through the IAGOS central database (http://iagos.sedoo.fr/#L4Place) and are part of the ancillary products (https://doi.org/10.25326/3) (Sauvage et al., 2017b, c). Each flight is archived separately and provides the mixing ratios of CO contributions from 14 anthropogenic and


biomass emitting regions throughout the world. The BB regions are obtained from the global fires data base (http://www.globalfiredata.org/data.html). Values larger than zero indicate anomalous mixing ratios exceeding the climatological background while null values are recorded when differences with the background are zero or negative. The modeled CO values were gridded in 0.25° x 0.25° cells for comparison with the *in situ* data.

The current study considers only observations within the -30° to +30° latitude band and from -50° to +180° in longitude (Fig. 1), excluding the high latitude flights in the northern hemisphere. This region encompasses the geographical region with most of the EIE in the dataset and it coincides with the climatological location of the inter-tropical convergence zone (ITCZ) as it moves between hemispheres following the seasons. Moreover, our analysis merges the 14 SOFT-IO regions into four larger ones: 1) Australia and Oceania (AO), 2) Persia, India and SE Asia (PIA), 3) Northern Hemisphere Africa and Atlantic Ocean (NHAA), and 4) Southern Hemisphere Africa and Atlantic Ocean (SHAA) as identified by the dashed boxes in Fig. 1. These regions not only encompass the majority of the EIE encountered but also correspond to regions of large sources of BB and UP emissions. The data are further stratified by rainy and dry seasons.

The data from the Aqua and Terra satellites' Moderate Resolution Imaging Spectroradiometers (MODIS) sensors are the Thermal Anomalies Level 2 Collection 6 data products (MYD14 and MOD14, Giglio et al., 2016) that provide the location of fires and their fire radiative power (FRP) at a spatial resolution of 1 km at nadir. The FRP (MW or MJ/s) was developed by Kaufman et al. (1998) from the 4 μm and 11 μm bands to determine fire intensity, which is linked to emissions. Monthly climatological maps are calculated from Aqua (nadir pass at 1:30pm LT) and Terra (nadir pass at 10:30am LT) over the period 2011-2019. Further, we use the aerosol optical depth (AOD) products from MOD04-Terra and MYD04-Aqua (Remer et al., 2009) at 550-nm wavelength and a spatial resolution of 10 km at nadir. The AOD are used to identify regions of high aerosol loading, in particular those that encompass regions of BB and UP. One caveat in the use of these data is that the AOD will not be available in regions co-located with extensive convection and clouds because of the cloud filtering algorithm used to generate the final data set.

The new generation reanalysis from the European Centre for Medium-Range Weather Forecasts (ECMWF), ERA5 (Hersbach et al., 2018) at 1° horizontal resolution is used to estimate climatological values of several meteorological variables such as upper-level horizontal wind divergence and outgoing longwave radiation (OLR). The wind divergence is related to upward vertical motions and the OLR is an independent indicator of cloud cover.

In addition, selected variables from the reanalysis MERRA-2 (The Modern-Era Retrospective Analysis for Research and Applications, Version 2) are analyzed over the same time period as the IAGOS measurements. MERRA-2, produced by NASA's Global Modeling and Assimilation Office (GMAO), is a multi-decadal global dataset that combines hyperspectral radiance, microwave data, GPS-Radio occultation data, ozone profile observations, and several other ground and satellite-based datasets (Randles et al., 2017; Gelaro et al., 2017). The horizontal resolution is 0.625° (longitude) x 0.5° (latitude). The M2I3NVAER product in 72 vertical levels is generated from the output

of the Goddard Chemistry Aerosol Radiation and Transport model (GOCART), which simulates an external mixture of 15 particle types. The importance of the use of the M2I3NVAER product in this study is that it provides a gridded output of speciated aerosol mixing ratios (Bocquet et al., 2015; Gelaro et al., 2017), i.e., dust, organic carbon and black carbon (OC, BC) and sulfate ($SO_4$). The mixing ratios of the different particle types in the layers between 8 and 12 km were averaged (weighted) and converted to standard temperature and pressure to determine

concentrations. Sea salt, another important aerosol type, especially over oceans, was excluded from the analysis for two reasons: 1) sea salt concentrations derived from the M2I3NVAER product, averaged over the same times and altitude layer, were two orders of magnitude lower than any of the other three aerosol types and 2) sea salt is an excellent CCN and when a water droplet forms on this particle type, it is already large enough to grow quickly to a size that collides and coalesces efficiently with other droplets to form raindrops that precipitate at altitudes lower

than where ice crystals are found. We acknowledge that this does not exclude sea salt as a potential source of ice crystals of liquid origin, but in comparison with the other three aerosol types, it will likely be less of a factor leading to the observed concentrations of ice in EIE.

Lighting flash counts and locations over the period 2011 to 2018 were obtained from the World Wide Lightning

Location Network (WWLLN) (Dowden et al., 2002) and were gridded to coincide with the MERRA-2 horizontal grid. The lightning flash count complements the OLR data, identifying regions of deep convection within the regions of widespread cloudiness.

### 3.0      Results

This section presents results as statistics derived from the nine years of accumulated measurements, first from only

the BCP cloud measurements, then only the CO in situ measurements accompanied by the anomalous CO extracted from the back trajectory and attribution model, and finally an evaluation of the cloud and CO data together.

### 3.1      Ice cloud statistics

The cumulative frequencies are calculated for ice crystal number concentrations grouped into each of the four regions mentioned above (Fig. 2a) and as a function of season (Fig. 2b). The distribution representing all clouds

(black curve) reaches 5000 $L^{-1}$ at the 85[th] percentile, the concentration that we are designating as the EIE threshold (indicated by the vertical dashed lines in Fig. 2). The NHAA and SHAA regions (orange and red curves) show similar frequency distributions beyond the EIE threshold, with 15% of the clouds in the EIE category. Those in the AO region (teal curve) have 25% of the clouds with concentrations > 5000 $L^{-1}$ while the PIA regions (blue curve) had 20% of the clouds with concentrations greater than the EIE. Differences in the frequency distributions as a

function of dry vs rainy seasons, independent of the regional areas, are slight (Fig. 2b), 13% vs 18%, respectively.

The ice number concentrations in the region of interest were mapped as a function of temperature similar to the cloud climatology presented by Krämer et al. (2020) in their Fig. 7a. Figure 3 shows the frequency of $N_{ice}$ as a function of temperature with the median and 90[th] percentile values drawn in yellow lines, solid and dashed,

respectively. Also drawn on this figure are the climatological values reported by Krämer et al (2020). Those data

from in situ observations are drawn in orange and from the satellite-derived DARDAR-$N_{ice}$ (Sourdeval et al, 2018) in magenta. The range in the IAGOS $N_{ice}$ is somewhat smaller than Krämer et al (2020), 0.05 – 100 cm$^{-3}$, vs 0.0001 – 1000, as is the temperature, 210-245 K vs 185 to 245. This is because the lower size range of the BCP is 5 µm whereas the instruments used in the Krämer et al (2020) studies were 2 µm. Nevertheless, the IAGOS data set from 335    2011-2020 is in very good agreement with the in situ climatology from the years 1999- 2017.

### 3.2    Carbon monoxide statistics

The CO anomalies, derived from the SOFT-IO model at cruising altitude, are separated into different categories considering the region, season and type of the source, i.e., all anomalies and those associated with BB or UP emissions, as shown in Fig. 4 with statistics listed in Supplemental material Table SM1. The data are grouped into 340    0.25° x 0.25° bins (28 km x 28 km) and the maximum BB and UP anomalies were calculated at each grid point. All of the gridded maxima are averaged to produce Fig. 4a. The average maximum CO anomalies due to BB are larger in the AO and PIA regions while those due to UP and BB are comparable in the NHAA and SHAA (Fig. 4a). These differences in averages, however, are not statistically significant because of the large standard deviations (vertical bars in Fig. 4a) Similarly, although the average CO anomalies appear to be twice as large during the rainy season for 345    the AO, PIA and SHAA, the standard deviations are also quite large (Fig. 4b).

Figure 5 highlights the frequency with which the CO anomalies measured over each region are coming from source regions different than the ones within which they were measured. The frequency of CO anomalies due to CO from BB sources is shown in Fig. 5a and from UP in Fig. 5b. The emissions from BB and UP in the PIA region dominate 350    the anomalies in CO over the AO and PIA (Figs. 5a, b).  Over the NHAA and SHAA regions it is the BB emissions from the NHAA that have the largest influence on CO anomalies encountered over the Africa and Atlantic regions (Fig 5a). The CO anomalies from UP sources, over the NHAA and SHAA, are primarily from surface sources in these same regions (Fig. 5b).

### 3.3    Combined carbon monoxide and ice cloud statistics

355

Figure 6 shows the locations of 1) CO anomalies that are not associated with ice clouds (black dots), 2) ice clouds with no CO anomalies (blue dots) and 3) clouds with CO anomalies (yellow dots, Fig. 6a and magenta dots in Fig. 6b). Figure 6a includes all ice clouds and Fig. 6b is with EIE only. With the exception of several flights to South America and Southern Africa, a large fraction of the flight routes within this region of the world have ice clouds not 360    only accompanied by CO anomalies, but also have EIE with the CO that is associated with air ascribed to surface sources. The black star markers are cities with populations larger than ten million, i.e., megacities (these are listed along with their locations and populations in the Supplementary Material, in Table SM3).

Figure 7 further clarifies the regional distribution of the average CO anomalies, for all ice clouds, separated by the 365    type of emission source, similar to how the CO data are presented in Fig. 4. The average anomalies, when associated with ice clouds, are similar to those with all CO, i.e., BB emissions exceed UP in the AO and PIA but there are no

differences with respect to these same sources in SHAA or NHAA. The largest anomalies of total CO, i.e., when both BB and UP are contributing, are over the PIA regions; however, the standard deviations (vertical bars in Fig. 7) indicate that some differences in the averages may not be statistically significant. Average values and standard deviations are listed in supplementary Table SM2.

Figure 8 summarizes the ice cloud events that are co-located with CO anomalies, similar to what is shown in Fig. 5, but differentiated by ice clouds with EIE (Fig. 8a) and non-EIE (Fig. 8b) and related to regions that are the source of the CO anomalies. The EIE in the AO and PIA regions are equally influenced by CO from the PIA, between 70 and 75%. A further 20% of the EIE contain CO anomalies (Other) that originate in regions located northward of $30^0$ N, beyond the current study's area of interest (Fig. 8a). Seventy percent of the CO in EIE over NHAA and SHAA originates from the surface under the same region where the clouds were encountered (Fig. 8b). The frequency distributions of the non-EIE with CO (Fig. 8b) have the same patterns as the EIE, with only slightly different magnitudes of the source contributions.

**4.0    Discussion**

Based on the results presented in the previous section, we can conclude that the SOFT-IO model attributes the sources of a large fraction of the CO encountered within the sampled ice clouds are biomass burning and urban emissions at the surface. We will now relate these emissions to the primary source regions and discuss how the CO produced in these areas is able to reach aircraft altitudes. This connection is an important key to supporting our conjecture that a significant number of ice clouds in the tropics and sub-tropics are formed on CCN or INP produced by BB and fossil fuel combustion in urban regions.

**4.1    CO surface sources and aerosol particle composition**

MODIS measurements (in Aqua and Terra) provide thermal anomalies that are used to derive the averaged FRP (in MW), over the years 2011-2019, associated with areas of enhanced emissions. We highlight here only the spatial distributions for July and December (Fig. 9), in which the magenta and blue colors correspond to the more intense fires. Also drawn on these maps are the EIE (black markers) from IAGOS flights observed during the same months. Figure SM2 in the supplementary material shows the spatial distribution of the monthly variability of the FRP illustrating the seasonality of biomass burning in different regions of the world. Note that in Africa, most of the emissions are from July through October in the southern hemisphere (Fig. SM2), while in the northern hemisphere, they predominate in November and December. Over Asia, in particular southeast Asia, the primary burning season is from July through October.

The AOD derived from the MODIS on the Aqua satellite measures the particle light extinction in a vertical column of air, which is a very good indicator of the particle mass in that column. The AOD spatial distribution, shown in Fig. 10, compared with the fire maps, CO anomalies and ice cloud distributions, provides independent observations of the regions of BB emissions. Note that the AOD is derived after filtering for clouds, so that only aerosol particles

are contributing to the light extinction. In those regions where clouds were frequently present during the period of interest, the averaged AOD values will often appear lower than expected. The values shown in Fig. 10 (and in all months shown Fig. SM3) have been normalized to the maximum value measured in the domain.

The SOFT-IO model attributes the CO anomalies encountered at cruising altitude to surface sources of the air that is mixed within the ice clouds (as seen in Figs. 5 and 8) but cannot provide any information about the particles co-emitted with CO, neither their composition nor their concentration. Nevertheless, the independent information of monthly mean AOD and FRP maps and especially the spatial distributions of aerosol composition mixing ratios, help corroborate the source regions identified by the CO back-trajectory analysis. The particle compositions relevant for this study are desert dust (Figs. 11a and b) and the sum of organic and black carbon (Figs. 11c and d) and sulfate as a indicators of urban sources (Fig. 11e and f). Only July and December are discussed here but the seasonality of the particle composition in the upper troposphere (at altitudes between 8 and 12 km) can be seen in the supplementary material (Figs. SM6-SM8).

A comparison of the AOD (Fig. 10) and the FRP maps for July and December (Fig. 9) shows that the higher AOD values are co-located with some of the same areas as high FRP in both seasons. High AOD values located over the Atlantic off the west coast of Africa, indicate aerosol particles that have been transported by easterly winds from the dust and BB sources. The lack of more one-to-one co-locations between AOD and EIE in this climatological perspective over nine years is a result of the underestimation of AOD in the presence of clouds. This becomes more obvious when we discuss the OLR and lightning maps (Fig. 12) that correspond to the high frequency of deep clouds in those regions where the AOD is much smaller than in those areas with less cloudiness.

Not only is northern Africa inundated with dust at flight levels, but the region over the Arabian Peninsula, the Indian Ocean, India and China also have dust concentrations in excess of 10 $\mu gm^{-3}$, coinciding with significant AOD. The EIE are also clustered in this dust belt band, particularly over and downwind of India. The presence of dust in the vicinity of EIE is significant because dust particles are very good INP (Cziczo et al., 2013; Demott et al, 2003). This suggests that perhaps the high ice concentrations could be more a result of the dust particles acting as INP than anthropogenic aerosol that are frozen water droplets

A large area of sulfate particles is seen in July stretching from western Africa to the Pacific Ocean, which are mixed with dust particles over Arabia, India and southern China, Thailand, Cambodia and Vietnam. These high concentrations of sulfate particles coincide with many EIE over southeastern Asia and recall also, that the SOFT-IO identified UP as the primary source of CO in the ice clouds over this area.

Some of the higher AOD values in Asia (Fig. 10b) may correspond to regions of BB in December across central and northwest India, supported by FRP emissions (Fig. 9b) and by the presence of organic and black carbon mass concentration at flight level from MERRA-2 reanalysis (Fig. 11b). However, high AOD values observed in July

over the Indian Ocean (Fig. 10a) and India are more likely related to dust and sulfate particles (associated with urban sources), as shown in Figs. 11a and 11e. The SOFT-IO model attributes the majority of CO anomalies observed at flight level within the Persian-India region with BB and urban sources within the same region (Fig. 5), as do both EIE and non-EIE co-located with CO anomalies (Fig. 8).


The region in southeastern Asia, over Indonesia and northern Australia (Fig. 10), is an illustrative example of how the cloud filtering may lead to an underestimate of the AOD. When compared with the OLR and lightning maps in Fig. 12, the widespread low OLR values and high frequency of lightning strikes in this region help understand the moderately low AOD in the same area. Sulfate plumes at flight level are seen in July over southeastern Asia,

particularly the region of Vietnam, Malaysia and Indonesia. The co-location of EIE, sulfate and FRP provides evidence supporting the results from the SOFT-IO model that attributes anomalous CO within EIE clouds to urban emissions (Fig. 5b) and BB emissions (Fig 5a) from within the same region.

The high AOD values in northern hemisphere Africa in July (Fig 10a) are not associated with BB as indicated by

FRP (Fig 9a) but rather with dust particles, supported independently by the aerosol composition shown in Fig. 11a, which indicate dust concentrations > 10 $\mu$g m$^{-3}$ in the upper troposphere. Dust particles are efficient ice nucleating particles (Cziczo et al., 2013; Demott et al, 2003), and EIE in the vicinity is likely associated with dust.

The SOFT-IO model attributes the majority of CO anomalies observed at flight level with BB (Fig 5a) and urban

sources (Fig. 5b) within northern and southern hemisphere Africa. Large regions of BB emissions are associated with regions of high FRP that straddle the Equator between December and July, contributing to high AOD, but most importantly, associated with high organic and black carbon mass concentrations (Figs. 11c and d).

The EIE co-located with CO anomalies are attributed by SOFT-IO to sources in Equatorial Africa (Fig 8.),

consistent with the areas of large FRP (Fig 9), and also with urban emissions within the same region. In December when there are EIE along the airline routes over the Atlantic Ocean (between northern Africa and South America), the CO back-trajectory analysis points to the region of enhanced BB emissions in northern Africa, supported by the spatial distribution of organic and black carbon mass concentrations, which indicates that BB is the dominant source in Equatorial Africa both in December (Fig. 12d) and in July (Fig. 12c). Nevertheless, the high concentrations of

sulfate particles indicative of urban emissions in July (Fig. 11e), which are mixed with dust over Africa and also contribute to high AOD, are consistent with the urban source attributed to the CO anomalies by SOFT-IO.

### 4.2    Identifying regions of vertical motions


The final piece of the ice cloud and EIE puzzle to be evaluated is the mechanism by which the CO and associated particles are transported to the UT where they are encountered by aircraft. Deep convection, forced orographic

lifting or frontal systems are mechanisms that can transport surface pollutants into the UT. The combined analysis of OLR, lightning and upper-air wind divergence (at 200 hPa) which is directly associated with underlying vertical

motions, provides evidence to support deep convection as the dominant pathway for anomalous CO concentrations at flight level. Figure 12 shows the mean spatial distributions of upper-level divergence, OLR, and lightning counts for July and December with EIE indicated by black markers. Low values of OLR are a proxy for the presence of clouds, indicating that the long wave emissions are dominated by cloud tops with very low temperature, characteristic of deep convection and ice clouds. On a monthly mean basis, upper-level divergence provides

information of the broad patterns associated with the presence of clouds indicated by the OLR distribution. Lightning counts are preferentially observed in deep convective clouds and squall lines in tropical regions and provide evidence of convection at the microscale. Corresponding maps for all 12 months can be found Figs. SM4 and SM5.

The large, upper-air divergence corresponding to the ITCZ over the Atlantic and Africa shifts from the northern hemisphere in July to well-south of the equator in December, when the largest number of EIE are encountered in the same region. Additional evidence for vertical motions can be inferred from the OLR maps (Figs. 12 c,d) and lightning count (Figs. 12 e,f). In Figs. 12 c,d over the Atlantic ITCZ, where there are many EIE, the upper-level divergence is associated to underlying vertical motions coinciding with low values of OLR and high AOD,

indicative of large mass concentrations of particles. Particularly significant is the cluster of EIE over this region of upper-level divergence in July, while virtually no such events exist in December, except directly over the western coast where the divergence is also large. The lack of EIE over tropical southern hemisphere Africa in December, even when there is strong upper-level divergence, and high lightning counts is due to the scarcity of flights in that region (Fig. SM1).


Large seasonal variability is observed over the Persia-India region associated with the southwest Indian monsoon in July, evidenced in all three variables: upper-air divergence, OLR and lightning, when which combined indicate generalized deep convection, corresponding with the higher frequency of EIE encountered in July.

There is less seasonal variation in spatial distribution over southeastern Asia where we observe appreciable divergence that corresponds to the locations of the EIE during July and December. Although the amount of sulfate is much more pronounced in July than in December, there is an increase in the OC/BC during December over this region as well as persistent dust and sulfate particles. Hence, with the widespread vertical motion, there is sufficient uplift of material from the surface to transport aerosols to flight levels in July and December. The regions of high

lightning counts are in close correspondence with the low OLR and large upper-level divergence, further reinforcing convective updrafts as the mechanism for CO transport and for high ice crystals from liquid origin at flight level. Note the large clusters of EIE located in the same regions of lightning high frequency, low OLR and elevated upper-level divergence.

**4.3    Selected case studies**

In this section we provide the results associated with several specific cases, so that the observed EIE can be directly linked to the emissions and deep convection rather than in a more statistical sense as presented in sections 4.1 and 4.2 above.  We have selected cases with maximum EIE in which the back-trajectory model attributed the sources of the CO at flight level within the same geographical region as the EIE was encountered, related to the results presented in Figs. 5 and 8 and discussed in sections 3.2 and 3.3, respectively.  In particular, we present here case studies that the SOFT-IO model attributes to sources in Southeast Asia and Equatorial Africa to complement the tropical Atlantic region discussed above in climatological terms.

**4.3.1 EIE in SE Asia attributed to BB in SE Asia: 14 September 2015**

The EIE observed on 14 September 2015 off the coast of Vietnam ($9.1545^0$N $105.335^0$E) had an ice crystal concentration of 17127 $L^{-1}$, more than triple the EIE threshold.  The rainy season in this region extends from May to November so the EIE was observed during its peak when there is widespread lightning activity (Fig. 13a), indicative of deep convection; moreover, the upper-level divergence (Fig. 13b) indicates a region of maximum divergence immediately to the east of the EIE. Those two variables are directly related to the presence of active and deep convection on the day of the event.  The back-trajectory determined from the SOFT-IO model indicates that the CO anomaly (74 ppb) co-located with the EIE was linked to BB emissions in Equatorial Asia (area shown in Fig. 1, turquoise dashed line), which climatologically shows large emissions associated with BB during September (Figure SM2). However, there were no fire events identified near the EIE on 14 September, and no AOD values were retrieved due to the limitation to detect AOD in the presence of clouds (Fig 13d).  Note that the Aqua satellite, on which the MODIS sensor used to detect FRP and AOD is mounted, does not have full coverage of the Earth's surface on a single day of measurements. The blank regions are those areas that were not covered that day. Nevertheless, a large area of intense biomass burning was observed in western Borneo (Fig. 13c) on that day and it is worth noticing that this BB is occurring during the peak of the rainy season. However, as seen on the lightning activity panel, no deep clouds were observed on that day in western Borneo. Low-level horizontal winds and convergence (not shown) were consistent with flow over the sea towards the coast of southern Vietnam where the EIE was recorded. While emissions from the western Borneo source may have modified cloud microphysics, Hien et al. (2019) report from two years of measurements in Vietnam that September is associated with BB sources. Back-trajectories indicated predominant flow from southern Thailand, Malaysia and Indonesia towards Vietnam, leading to an increase in surface PM during the rainy season. Our case study is consistent with their results that were reported on a monthly basis.

**4.3.2 EIE in Equatorial Africa attributed to BB in Equatorial Africa: 19 June 2018**

An EIE with concentration of 61727 per $L^{-1}$ (more than an order of magnitude above the EIE threshold) was observed in western Africa ($9.6276^0$N, $0.8983^0$W) on 19 June 2018, indicated by the black dot in Fig 14. Lightning activity highlights the presence of areas of deep convection between $5^0$N and $15^0$N, and the EIE was observed at the

eastern edge of one of those areas. It should be noted that mid-June, when this EIE was observed, corresponds to a break in the monsoonal precipitation which starts in western Africa in March at the Equator and proceeds northward as it increases in precipitation intensity reaching a maximum around ~4-5N between May and early-June, as seen in Berthou et al (2019).

As in the southeast Asia case, deep convection is associated with broad areas of upper-level divergence (Fig. 14b). Note that there are no BB sources located in the region where the EIE was observed (Fig. 14c), but locally produced BB smoke is present (Fig. 14 d). However, the SOFT-IO model attributes the CO anomaly (14 ppb), co-located with the EIE observed on 19 June 2018, with a BB source in Equatorial Africa. Such an area of BB emissions, represented by the FRP, can be seen in Fig. 14c, centered in ~$5^0$S, $15^0$E with the corresponding aerosol loading (Fig. 14d). Low-level cross-equatorial monsoon flow will transport the emissions from these fire sources and subsequently influence cloud formation and evolution.

### 4.3.3 EIE in SE Asia attributed to UP in SE Asia: 21 October 2016

The EIE observed on 21 October 2016 off the west coast of Malaysia and at the latitude of Kuala Lumpur ($3.5304^0$N, $105.2796^0$E), had an ice crystal concentration of 108655 per $L^{-1}$ (about 20 times the EIE threshold). Climatological rains occur at this latitude throughout the year, with two periods of higher accumulated amounts: March-April and September-December; the EIE occurred in the middle of the second period. The spatial distribution of lightning counts observed on 21 October 2016 (Fig 15a) indicates generalized convective activity encompassing most of peninsular Malaysia, including over the sea just offshore of the east coast, where the EIE was encountered. The location coincides with large upper-level divergence (Fig 15b) indicative of deep convection. Note that there are no sources of BB detected nor AOD-valid values in the region, due to the presence of clouds. The SOFT-IO model attributes the CO anomaly (28 ppb), co-located with the EIE, to an urban source within the same SE Asia region. Notably, Kuala Lumpur, the capital of Malaysia and home to a population of 8 million, is located directly to the west of the EIE and is the most likely source of the CO anomaly measured at cruising altitude.

### 4.3.4 EIE in Equatorial Africa attributed to UP in Equatorial Africa: 9 August 2018

An EIE with concentration of 56894 per $L^{-1}$ (about an order of magnitude above the EIE threshold) was observed in western Africa (7.2757°N, 7.4541°E) on 9 August 2018. August is climatologically the middle of the rainy season at the northern-most latitudinal extent, before the ICTZ starts its movement equatorward. The spatial distribution of lightning counts (Fig 16a) indicates a large region of deep convection around 12N, oriented E-W, directly related to the ITCZ. Activity can be seen to the south of the ITCZ on this day, covering most of Nigeria where the EIE was recorded. The lightning pattern indicates deep convection associated with the large upper-level divergence (Fig 16b), which is particularly strong off-shore of the Niger river delta. There are only a few isolated sources of BB indicated in the FRP distribution (Fig 16c) near the EIE, with more widespread BB emissions to the southeast, centered between 5S and 20W. The generalized convection to the north has possibly scavenged the desert dust and no valid AOD (Fig 16d) is detected near the location of the EIE. The SOFT-IO model attributes the CO anomaly (19

ppb) to an urban source within the same region. Note that while the EIE is located close to Lagos, the megacity capital of Nigeria (population: 13.5 million), it is even closer to the extensive oil production fields in the vast delta of the Niger river, where venting and open burning of fossil fuel is carried out. This large anthropogenic source is most likely the source of the CO anomaly measured at cruising altitude.

### 4.4    Relationship with previous studies

The results that we have presented provide a framework for linking ice clouds in general, and EIE in particular, to surface sources of dust, BB and UP in tropical latitudes. Given that this study lacks *in situ* measurements of the actual ice crystal residuals, we are unable to unequivocally conclude that extreme ice events are due to emissions from BB and UP. Instead, we will place our results in the context of other studies of ice clouds in regions similar to those we have reported here. Krämer et al (2020) provide a comprehensive summary of the microphysical properties of cirrus clouds related to where they are found and to the aerosols from which they form. The ice clouds measured by the BCP in the latitude band of ±30° are most likely those that have been designated as liquid in origin, as previously discussed in the introduction, and given the presence of lightning where the EIE where recorded. According to Krämer et al. (2016, 2020), Luebke et al. (2016) and Costa et al. (2016), liquid-origin ice clouds are usually composed of high concentrations of water droplets that froze either homogeneously or heterogeneously at a lower altitude. Although some of the ice might have formed on INP, the greater abundance of CCN-type aerosol, relative to INP-type aerosol, would suggest that the ice formation process is through droplet freezing. Many laboratory, cloud chamber and field studies have shown that some BB and UP aerosol are hygroscopic and can serve as CCN, while fewer studies have also demonstrated their efficiency as INPs. Nevertheless, the liquid-origin clouds that formed in the tropics, as discussed by Krämer et al. (2016, 2020), were related primarily to deep convection and anvil outflow, similar to what we have reported here, where strong updrafts will lead to activation of particles regardless of their composition.

### 4.5    Relationship of EIE to ice crystal icing hazards

Ice clouds, in addition to having a major influence on climate through their interactions with incoming and outgoing radiation, are also a potential hazard to commercial aviation as a result of the clogging of air data sensors or by being ingested into engines, a condition that can lead to engine flameout or rollback (e.g., Mason et al., 2006; Grzych and Mason, 2009; Mason and Grzych, 2011; Bravin, et al., 2015; Haggerty et al., 2019; Bravin and Strapp, 2019). Figure 17 superimposes all of the EIE determined in this study from 2011 to 2019, onto the map of ice crystal icing (ICI) events reported by pilots from 1990 to 2019 (Bravin and Strapp, 2019). The map shown in Fig. 17, with the ICI events marked by the blue filled circles, is reprinted with permission from the authors and the Society of Automotive Engineers (SAE) International Journal on Advances & Current Practices in Mobility.

The juxtaposition of nearly all of the ICI events with the EIE underscores the importance of the link between hazardous flying conditions and the presence of ice clouds with extremely high crystal concentrations, clouds that in this current study have been associated with surface emissions of anthropogenic CO (through the SOFT-IO model)

and aerosol particles. To further underscore how the EIE is associated with ICI, in Figure 18 we show the relationship between $N_{ice}$ and IWC (Fig. 18a) and the cumulative frequency distribution of IWC derived from the BCP size distribution (Fig. 18b). The research community that investigates the impact of high IWC on aircraft performance uses 1 $gm^{-3}$ as a threshold for defining the IWC that they consider hazardous for aircraft operations (deLaat et al, 2017). Figure 18a shows that a large fraction of EIE far exceeds this threshold, keeping in mind the ±50% uncertainty in deriving IWC from the BCP. In addition, Fig. 18b shows that approximately 5% of all the ice clouds measured had IWC > 1 $gm^{-3}$, i.e., more than 2000 of the clouds encountered by commercial flights between 2011 and 2019 were potentially hazardous for aircraft operations.

### 5.0    Conclusions and recommendations

Measurements made with Backscatter Cloud Probes flown on eight Airbus 330/340 passenger airliners from December 2011 to March 2020, as part of the In-Service Aircraft for a Global Observing System (IAGOS), were analyzed to assess the properties of ice clouds in a region bounded by ±30° latitude and -50° to 180° longitude. A total of 67172 ice clouds were sampled during 22857 flights, representing 1483 hours in cloud. Approximately 20% of these clouds (6490) had number concentrations larger than 5000 $L^{-1}$, a concentration defined here as an extreme ice event (EIE).

Many of the flights also included measurements of carbon monoxide (CO), which can be used to assess the potential influence of surface-based, anthropogenic emissions on clouds at cruising altitude. The SOFT-IO model used in our analysis incorporates back trajectory analysis and CO emission inventories to attribute anomalous levels of CO sampled by the aircraft to surface emissions. The CO anomalies, values that exceeded a background concentration at flight level, were averaged into 0.25° by 0.25° grid cells and compared with ice crystal concentrations averaged into the same-sized cells. Not all CO anomalies were found in cloud and not all clouds had CO anomalies. Of all the cells that had either clouds or CO anomalies, 52% had CO anomalies with no clouds, 10% had clouds with no CO anomalies, and 38% had concurrent observations of clouds and CO anomalies. When only EIE were evaluated, the numbers were 75%, 4% and 21%, respectively. In other words, when considering all clouds, 79% were accompanied by positive CO anomalies while if only including EIE, 84% of these had anomalous concentrations of CO. Hence, while there is nothing that would suggest that EIE clouds are being influenced significantly more often than non-EIE clouds, anthropogenic emissions from BB and UP are associated with ice clouds a large percentage of the time in this latitude band..

The modal frequency of the temperature of the EIE clouds was five degrees colder than non-EIE clouds; however, this difference is insignificant when taking the temperature variance into account. Likewise, neither the differences in EIE over the study region or the differences in the magnitude of CO anomalies between EIE and non-EIE clouds were significant. Nevertheless, inspection of selected case studies with ice crystal concentrations much larger than the EIE threshold of 5000 $L^{-1}$, provides indications that the surface anthropogenic sources located in the same regions as the EIE, as attributed by the SOFT-IO model, may be contributing CCN (or even INPs) that would have intervened in cloud formation and crystal development.  The cases presented indicate that the EIE were co-located

with deep convection but not necessarily the regions of the strongest convection, as characterized by lightning counts (at the microscale) or upper-level divergence (at the mesoscale).

We conclude that the EIE encountered at cruising altitude in ice clouds were always associated with deep convection in the tropical band, and the co-located CO was attributed to specific sources by the SOFT-IO model.   Deep convection was characterized by the presence of lightning, upper-level divergence and OLR. The attributed sources were associated with large scale, climatological spatial patterns of fire radiative power as a measure of BB emissions and AOD as a measure of atmospheric aerosol loading. More detailed spatial distributions of dust, OC/BC and

sulfate concentrations, determined by the GEOS-Chem modelling and incorporated into the MERRA-2 database, provide climatological support of the presence of surface sources on a monthly basis. The case study analysis, on a daily basis, highlights the presence of EIE with concentrations much higher than the threshold and can provide support for the attribution obtained from the SOFT-IO model. We also conclude that the primary role of BB and UP emissions is the production of copious numbers of particles that can be activated as water droplets, regardless of

their composition, if lofted in vigorous updraft that lead to high supersaturations.

A qualitative comparison of regions with ice crystal icing (ICI) events and EIE shows an unmistakable spatial correspondence that underscores the potential importance of EIE on commercial aviation. The IAGOS program encountered 6490 EIE in a ten-year period, or approximately 650 per year. This seems small compared to the 22857

IAGOS flights during this same period, i.e., 2285 flights yearly, yet this implies that almost 30% of the flights encountered an EIE, assuming that the data are from only a single EIE encountered per flight. The significance of this number becomes clearer when taking into account the more than a million international flights made per year when potentially more than 300,000 of those flights might eventually encounter EIE. This statistic, while lacking robust validation, is sufficiently alarming to highlight the need for better forecasting of ice clouds along flight

corridors, in general, and the probability of EIE, in particular.

The results that we have presented here have shown how anthropogenic emissions, as well as desert dust, are lofted by strong convergence at the surface and UT divergence, leading to the formation of ice clouds and EIE, which in turn may also be responsible for the ICI. These results explain the prevalence of ICI and EIE in the tropics around

the ITCZ, especially over Africa and Asia, which is highlighted in the BCP measurements compared with pilot results (Fig. 17). More recently, Rugg et al. (2021) analyzed 11 years of Cloudsat and CALIOP measurements and showed that, on average, more than 2% of the UT clouds in this region contain ice water content > 1.0 gm$^{-3}$. Hence, a better knowledge of the sources of aerosol particles, their composition and their concentrations, will provide valuable information that can improve flight planning and the forecast of potential areas of EIE.


The current study has established a pathway that associates extreme ice concentrations with the surface emissions of particles from biomass burning and fossil fuel combustion; however, several unresolved questions remain:

- What is the relationship between ice cloud CO and the properties of the CCN or INP from which the crystals formed?
- What is the relationship between the age of the CO and the ice crystal concentrations, and in particular, what leads to the evolution of EIE clouds?
- What additional information is needed to accurately forecast ice clouds in general, and EIE in particular, not only for improved flight safety, but for a better understanding of the climate impact of these clouds?
- Does deep convection with strong updrafts minimize the importance of aerosol composition?

In summary, the results presented in this study emphasize the importance of programs like IAGOS for expanding the data base of cloud measurements and the need to broaden such programs to include more aircraft to provide a more complete spatial and temporal climatology.

**6.0    Data availability**

The BCP, CO and SOFT-IO CO trajectories are all downloadable, free of charge, from the IAGOS web site http://www.iagos-data.fr/. The AOD and fire radiative power are standard products from the Moderate Resolution Imaging Spectroradiometers (MODIS) sensors on the Aqua and Terra satellites and are available free of charge, from https://modis.gsfc.nasa.gov/. The data used to generate the maps of upper air divergence and OLR used the new generation reanalysis from the European Centre for Medium-Range Weather Forecasts (ECMWF), ERA5
(ecmwf.int/en/forecasts/datasets/reanalysis-datasets/era5) and the data used to create the maps of aerosol composition was downloaded from the MERRA-2, produced by NASA's Global Modeling and Assimilation Office (https://gmao.gsfc.nasa.gov/). Non-real time lightning data is available from the WWLLN website (http://wwlln.net/) at no cost for users who supply data to this network; otherwise, there is a charge for the data.

**7.0    Author contributions**

Graciela Raga and D. Baumgardner designed the study, interpreted the combined results and wrote the text, D. Baumgardner analyzed the BCP and CO measurements, B. Rios provided the analysis and associated figures from the Terra and Aqua MODIS data, Y. Díaz-Esteban extracted the information from the ERA5 reanalysis and generated the upper-level divergence maps, A. Jaramillo evaluated the MERRA-2 and WWLLN data and generated the associated maps of aerosol composition and lightning frequency, M. Gallager and G. Lloyd supplied the BCP
files and B. Sauvage and P. Wolff assisted with the interpretation and analysis of the SOFT-IO trajectories.

**8.0    Acknowledgements**
IAGOS gratefully acknowledges financial support during its preparation, implementation and operation phase from the European Commission in FP6 and FP7 programmes, national research programmes in Germany (BMBF), France (INSU-CNRS, MESR, CNES) and UK (NERC), in addition to institutional resources in Germany (Helmholtz
Association, Max-Planck-Society, Leibniz Association), France (Universite de Toulouse, Meteo-France) and the UK (University of Manchester, University of Cambridge), and the continuing support by participating airlines (Deutsche

Lufthansa, Air-France, Iberia in Europe, China Airlines and Cathay Pacific in Asia). The authors also thank the World Wide Lightning Location Network (http://wwlln.net), a collaboration among over 50 universities and institutions, for providing the lightning location data used in this paper and gratefully acknowledge the European

Centre for Medium-Range Weather Forecasts (ECMWF) and the Earth Observing System Data and Information System (EOSDIS) at NASA for data availability. Finally, we wish to thank the three, anonymous reviewers who provide many valuable comments and recommendations that greatly strengthen the scientific foundations of this study.

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

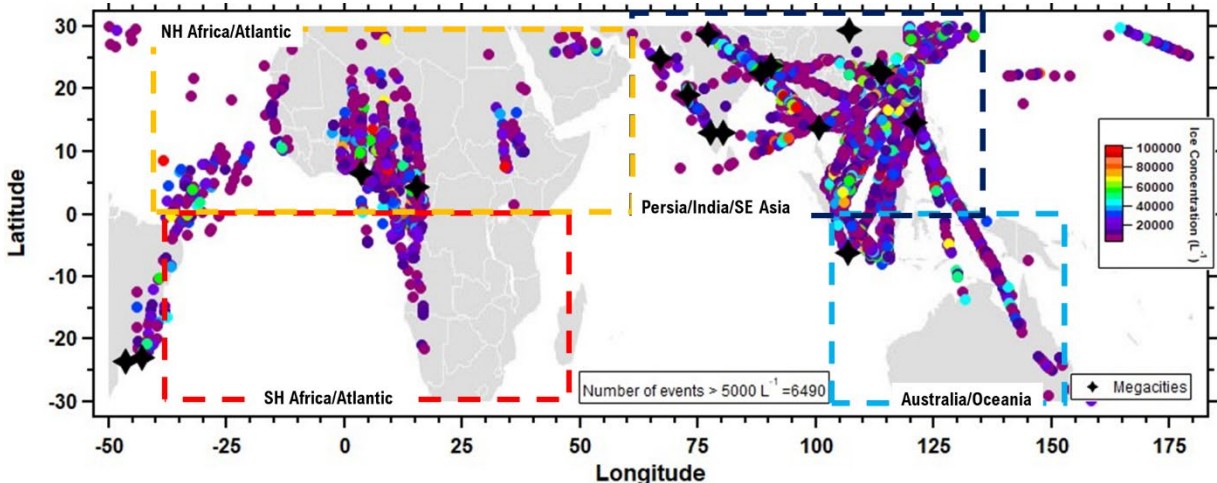

*Figure 1. Distribution of extreme ice events (EIE), at low latitudes, when ice cloud concentrations exceed 5000 L⁻¹ (2011-2020). The color scale indicates the ice crystal concentration ($L^{-1}$). The black stars indicate the location of megacities. The boxes with colored dashed lines delineate the regions into which the ice clouds are classified for*

*further analysis.*

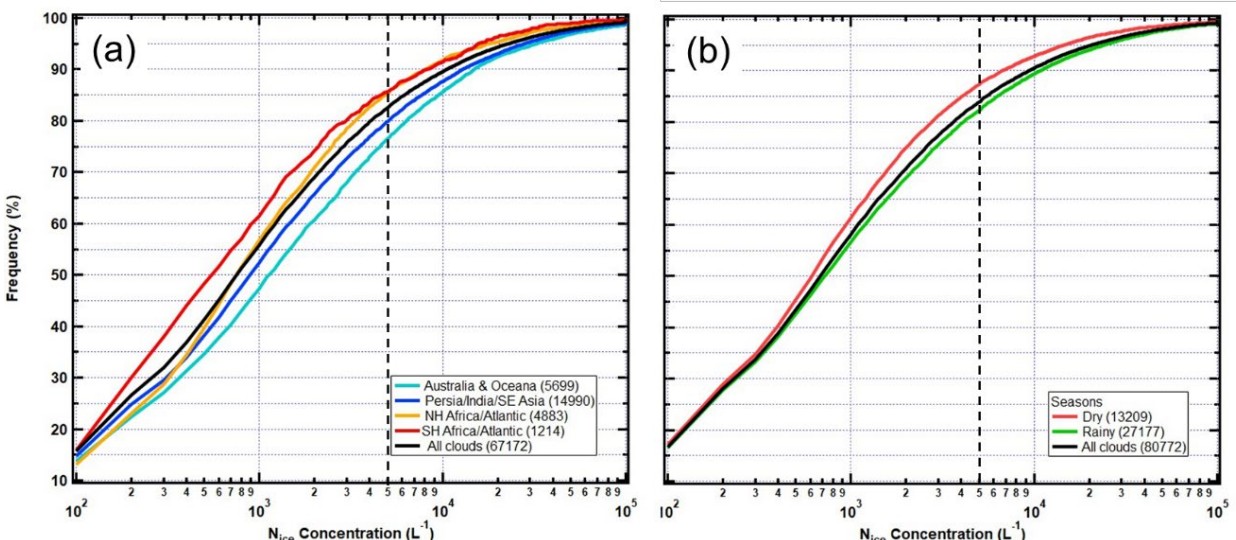

*Figure 2. Frequency distribution of $N_{ice}$ for all measured ice crystal clouds a) by region and b) by dry/rainy season.*

*The vertical dashed lines indicate the Extreme Ice Events (EIE) threshold concentration.*

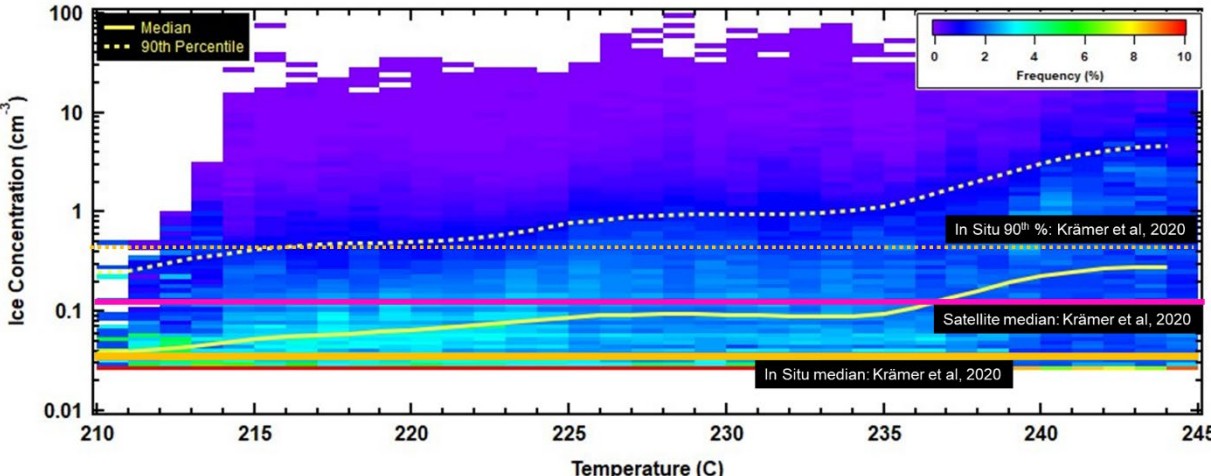

*Figure 3.  Frequency distribution of cloud ice concentration versus temperatures for all measured ice clouds. The measurements are presented in a format similar to Fig. 7b in Krämer et al. (2020) with the in situ median, satellite derived median and 90th percentiles shown by the horizontal orange, magenta and dashed orange lines, respectively. The median and 90th percentile values for the data set reported here are shown with the solid and dashed yellow curves, respectively.*

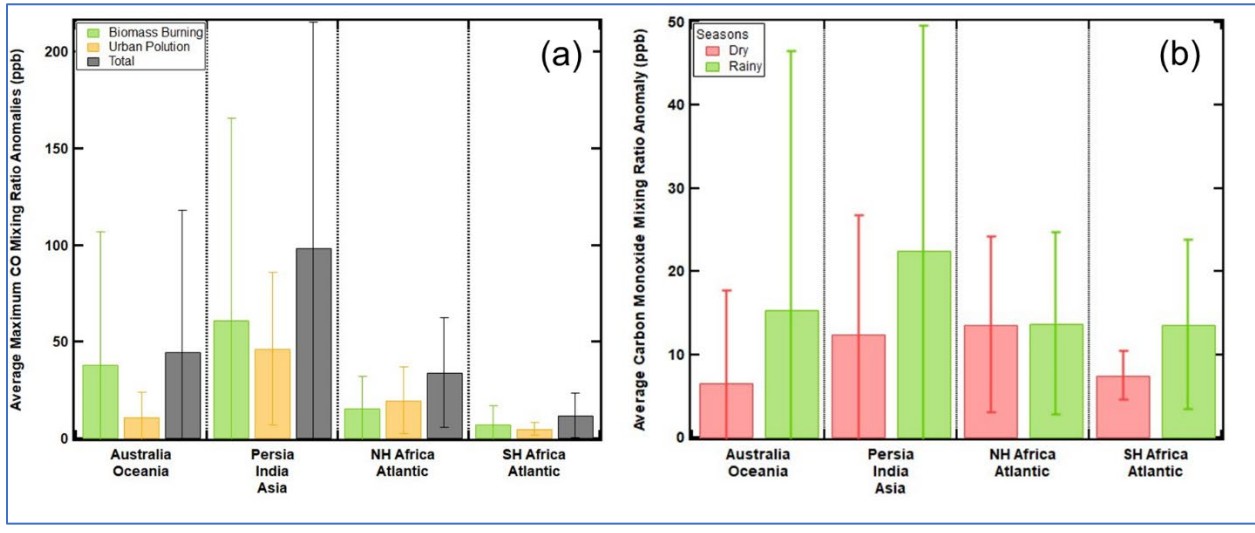

*Figure 4. The average CO anomaly as a function of geographical region and (a) source of CO and (b) season of the year. The vertical lines are standard deviations about the mean.*

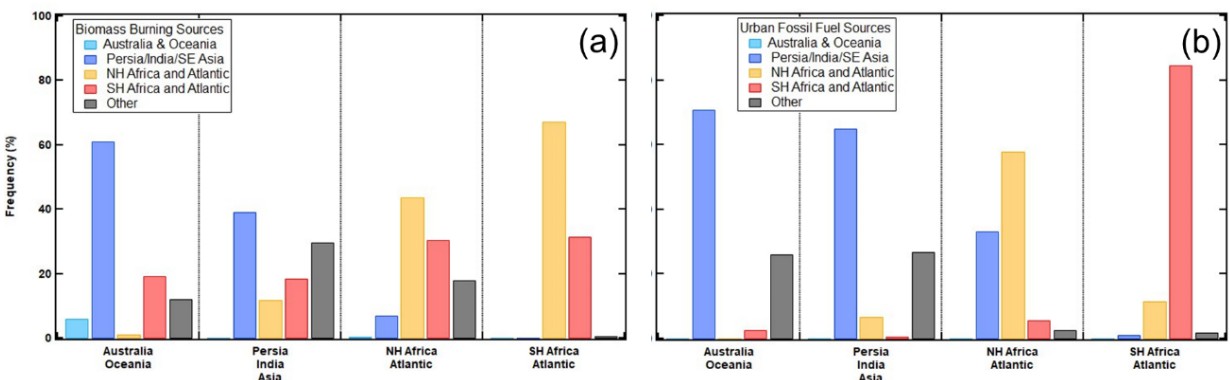

*Figure 5. The percentage of CO anomalies from a) biomass burning and b) urban pollution are shown by the source of the CO (colored bars) within each geographical region.*

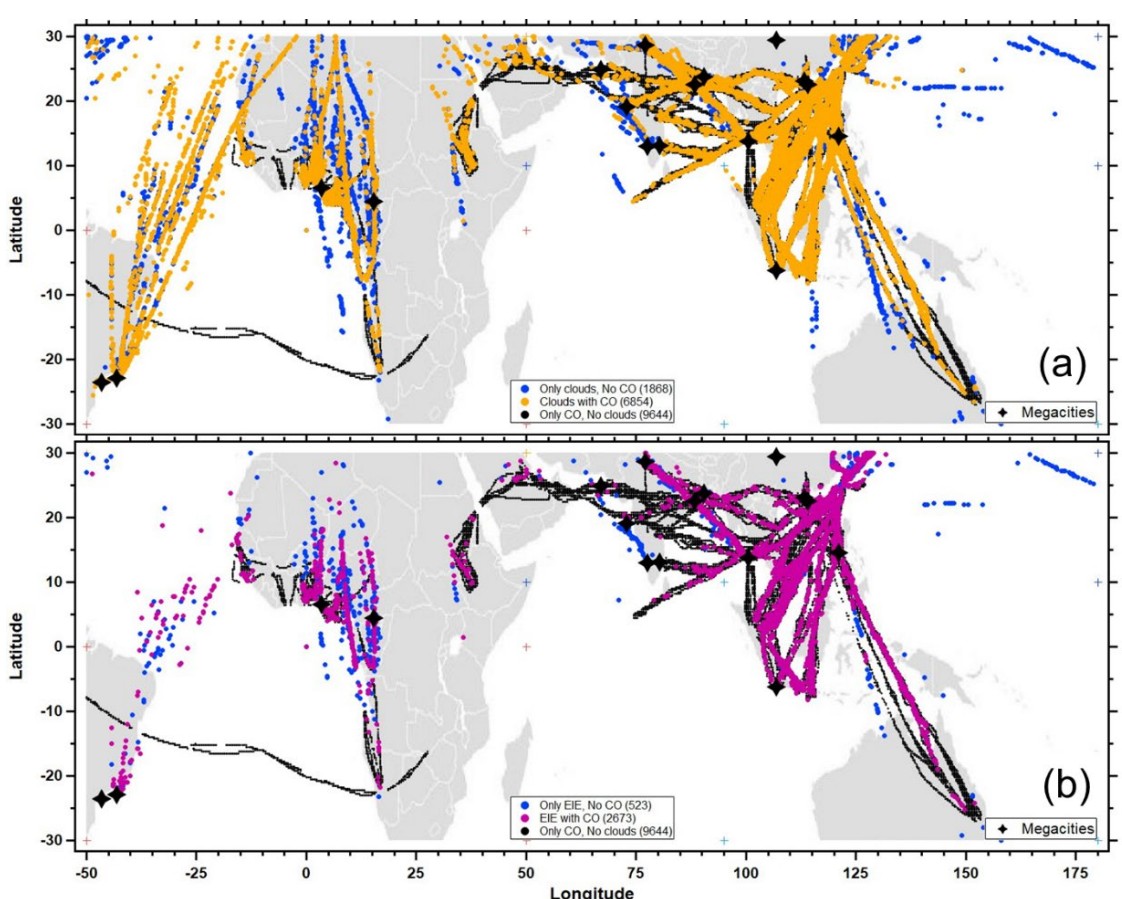

*Figure 6.  Carbon monoxide (CO) concentration anomalies (color scale) co-located with a) all ice clouds and b) only clouds with Extreme Ice Events (EIE). The back star symbols indicate the location of megacities.*

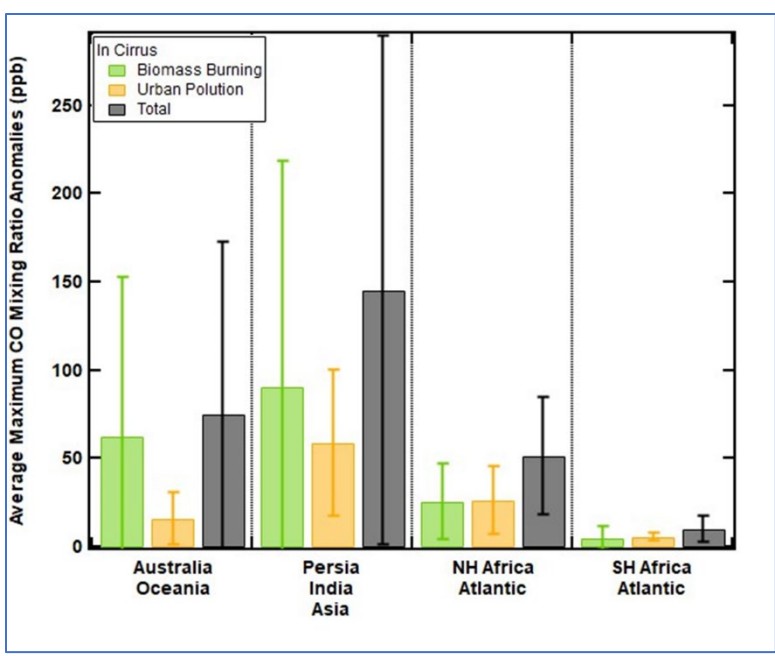

*Figure 7. Similar to Fig 4 but only for CO anomalies co-located with ice clouds.*

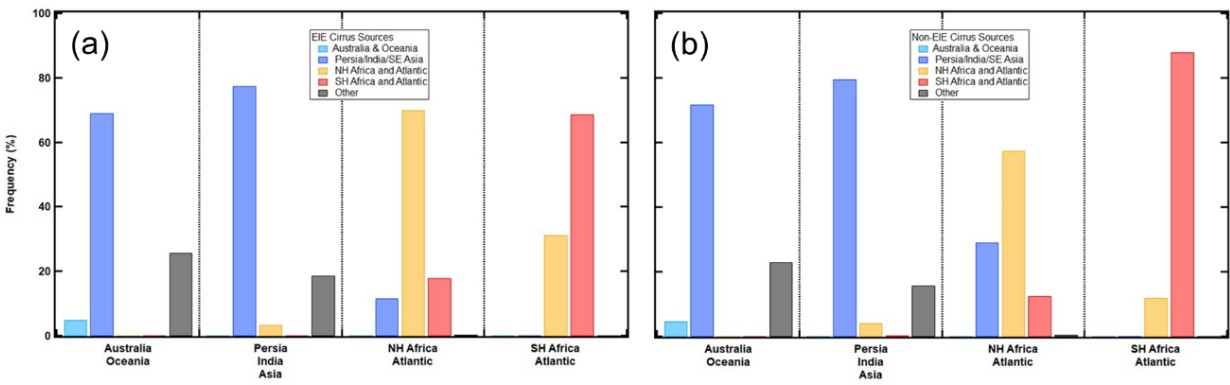

*Figure 8. Similar to Fig. 5 but a) the percentage of CO anomalies under EIE ice cloud conditions and b) non-EIE shown by the source region of the CO (colored bars) within each geographical region.*

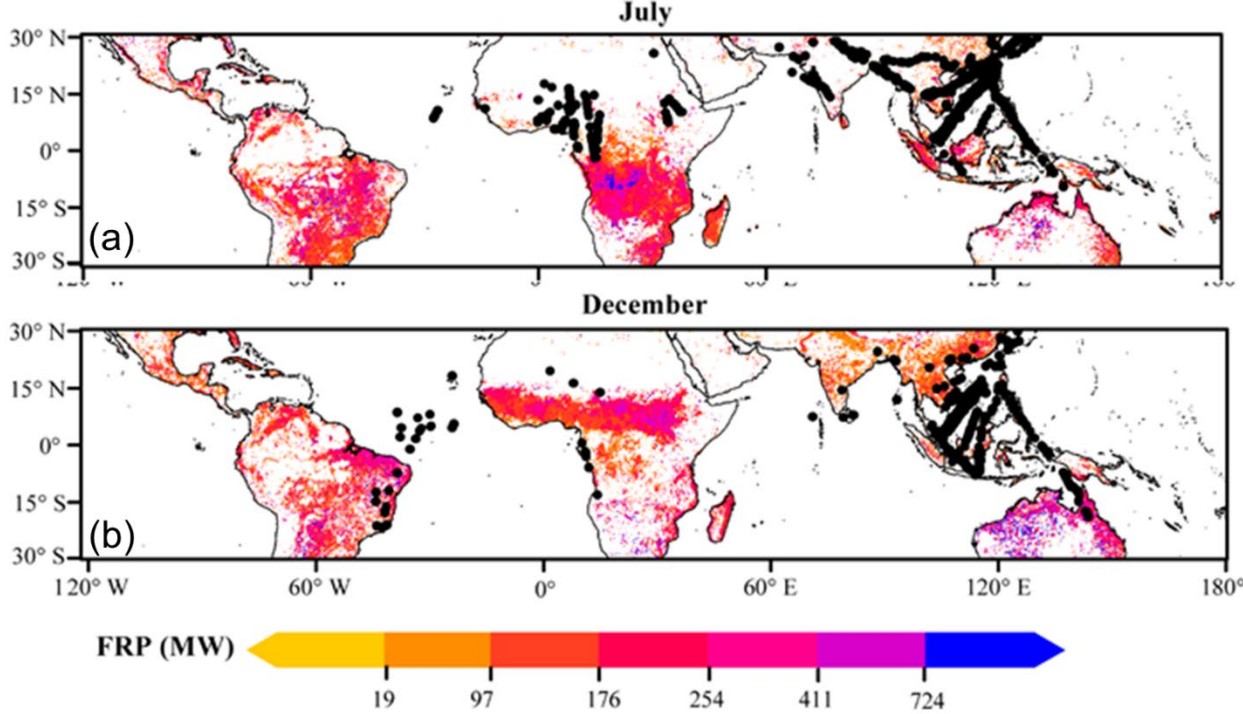

*Figure 9. Spatial distribution of the fire radiative power (FRP) derived from MODIS on the Terra satellite (nadir pass at 1:30pm LT) for (a) July and (b) December (2011-2019). The black markers show the location of EIE observed in each month.*

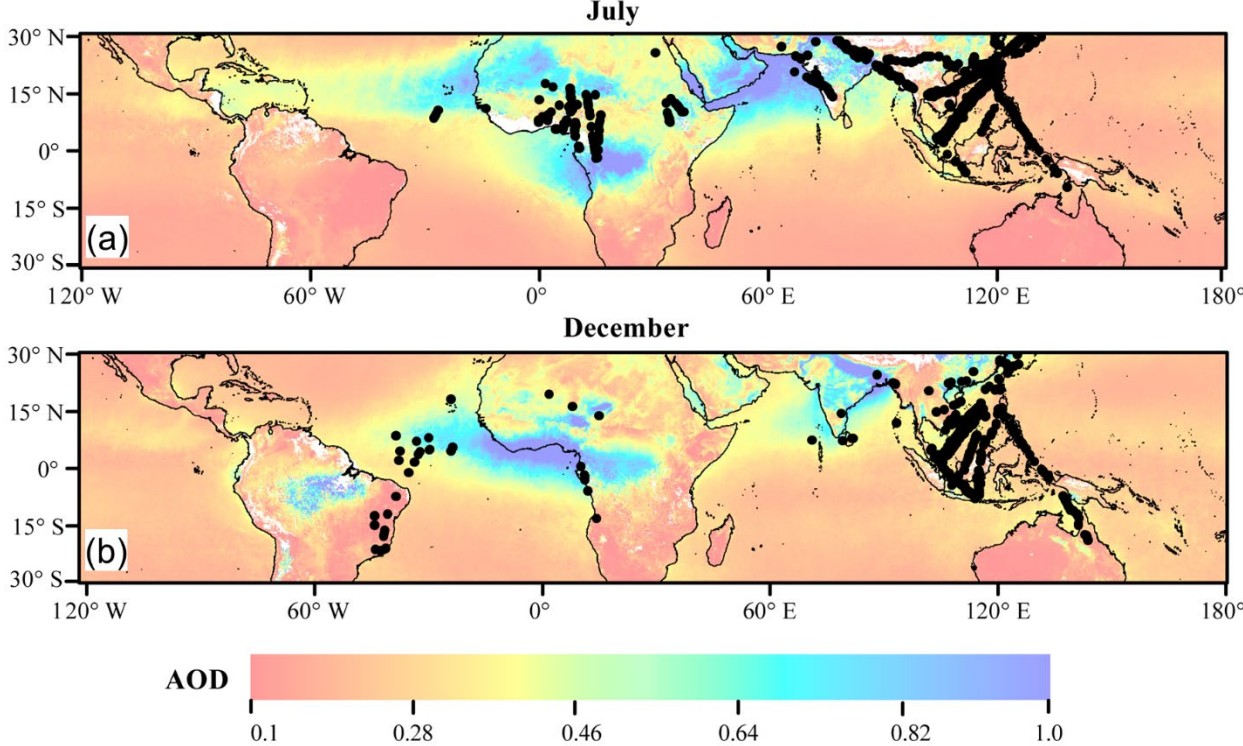

*Figure 10.  Spatial distribution of the aerosol optical depth (AOD) derived from MODIS on the Aqua satellite (nadir pass at 1:30pm LT), 2011-2019 for a) July and b) December. The EIE for July and December are shown as black markers. The AOD has been normalized so that the maximum value is unity. The few white regions correspond to grid points with very high fraction of clouds so that a valid retrieval of AOD is not possible.*



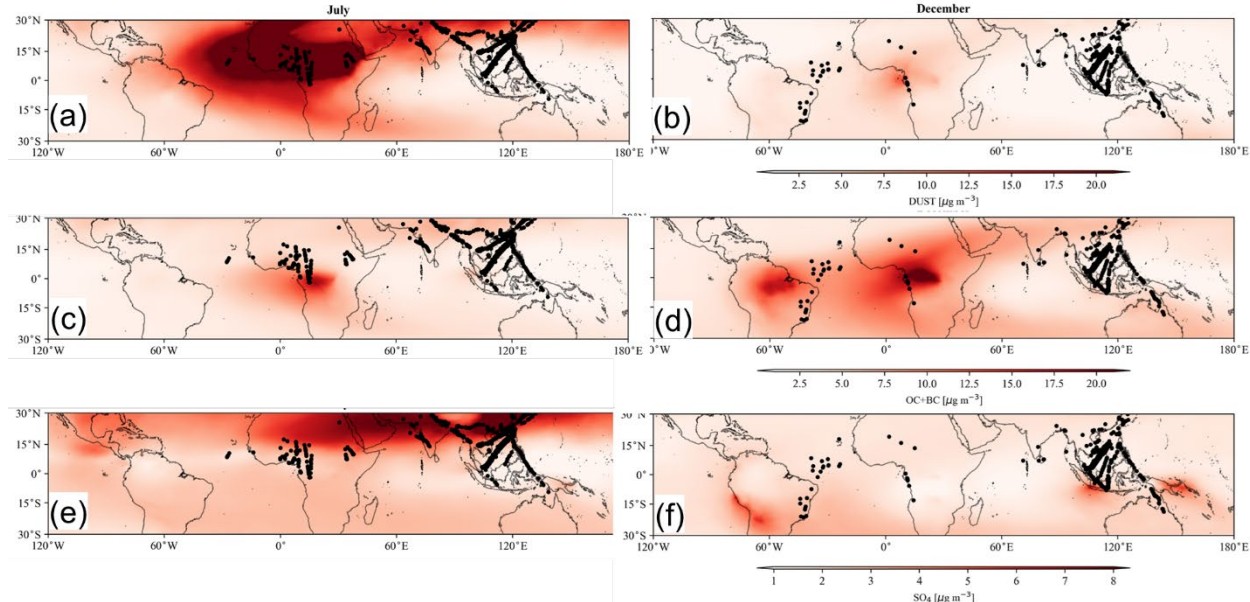

*Figure 11. Average particle mass concentrations by aerosol composition, for July (left panels) and December (right panels), derived from MERRA-2 for: a,b) dust, c,d) organic and black carbon (OC+BC) and e,f) sulfate. These are average concentrations between 8 and 12 km adjusted to standard temperature and pressure. The EIE for July and*

985                                      *December are shown as black markers.*

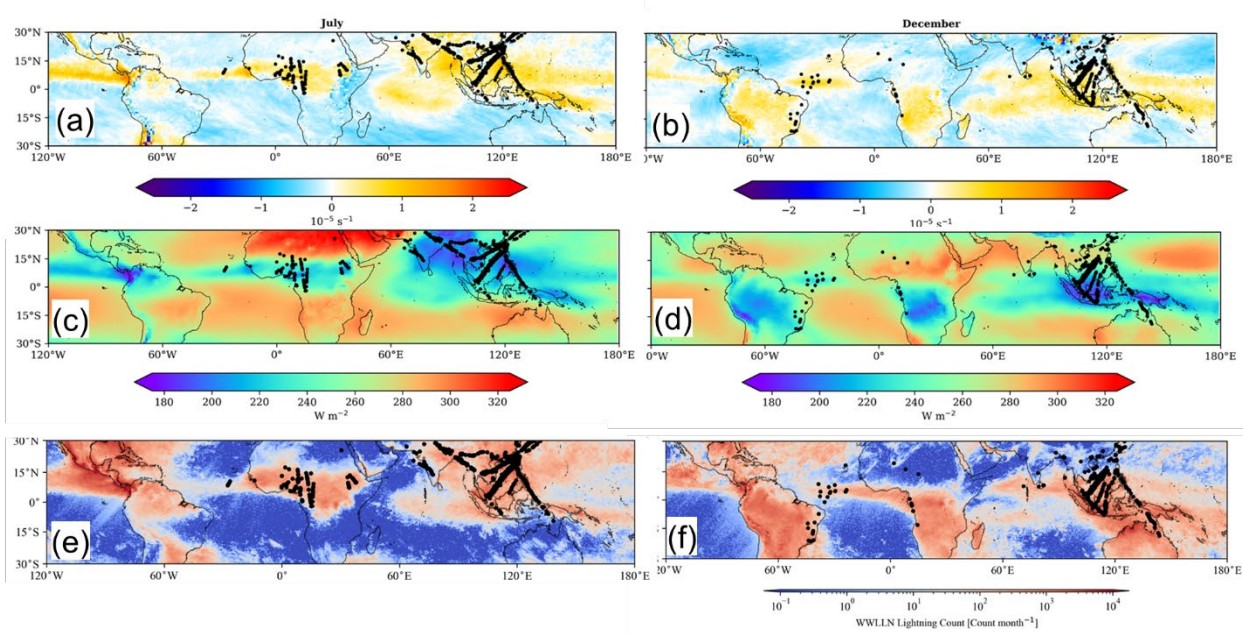

*Figure 12. Spatial distribution of the monthly average of horizontal wind divergence at 200 hPa, determined from ERA5 reanalysis (2011-2019) during a) July and b) December. Spatial distribution of the monthly average of outgoing long wave radiance for c) July and d) December. Spatial distribution of the monthly lightning flash count, derived from the World Wide Lightning Location Network (2011-2018) for e) July and f) December. The EIE for July and December are shown as black markers.*


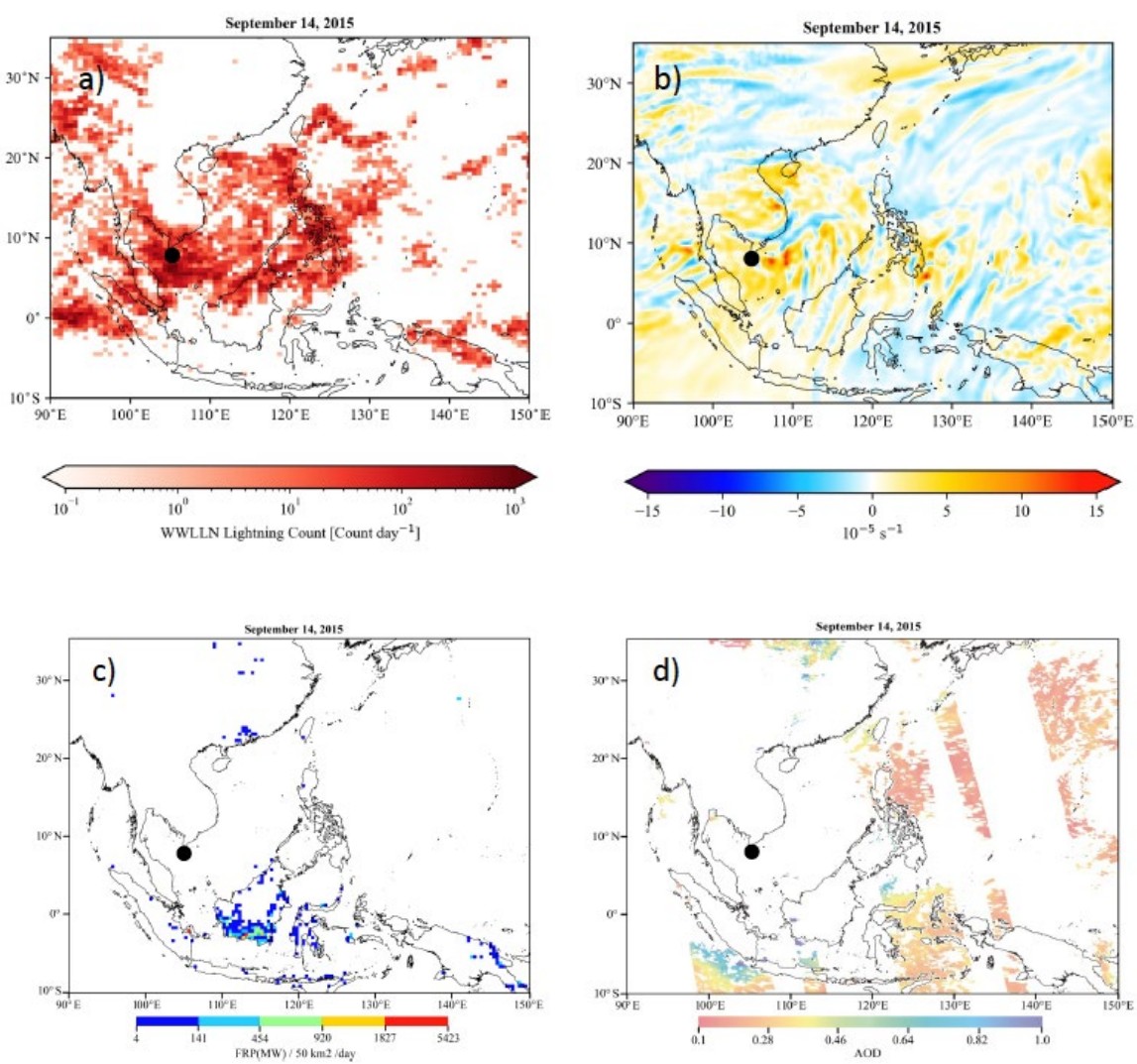

*Figure 13. Spatial distribution of a) lightning counts per day from the WWLLN database, b) horizontal wind divergence at 200 hPa from the ERA5 reanalysis, c) Fire Radiative Power (FRP) density per 5 km² per day from MODIS and d) the aerosol optical depth (AOD) derived from the MODIS sensor in the Aqua satellite (nadir pass at 1:30pm LT), for the EIE event recorded on 14 September 2015 (black dot)*


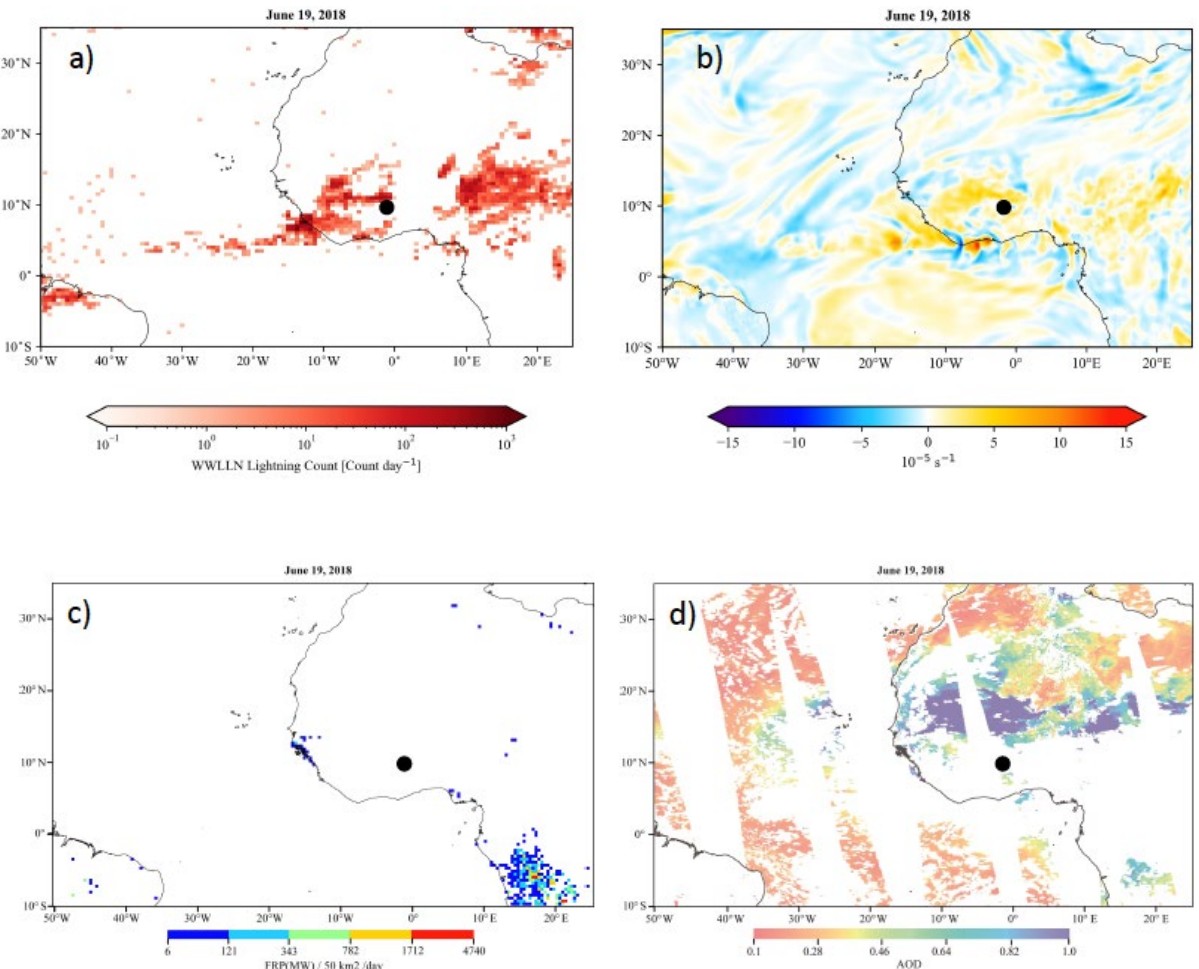

*Figure 14. Same as Fig. 13 but for the EIE event recorded on 19 June 2018 (black dot)*

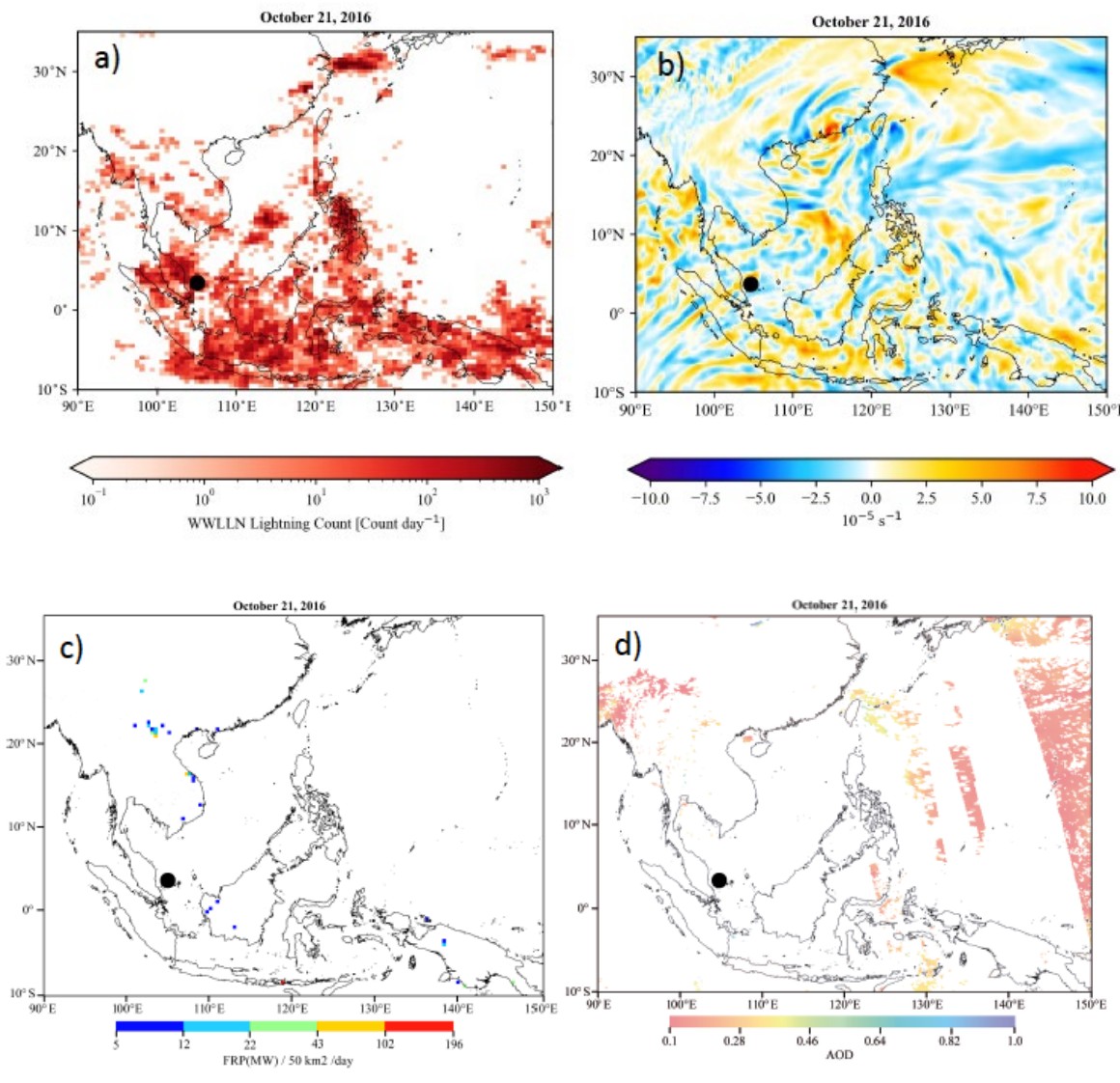

*Figure 15. Same as Fig. 13 but for the EIE event recorded on 21 October 2016 (black dot).*


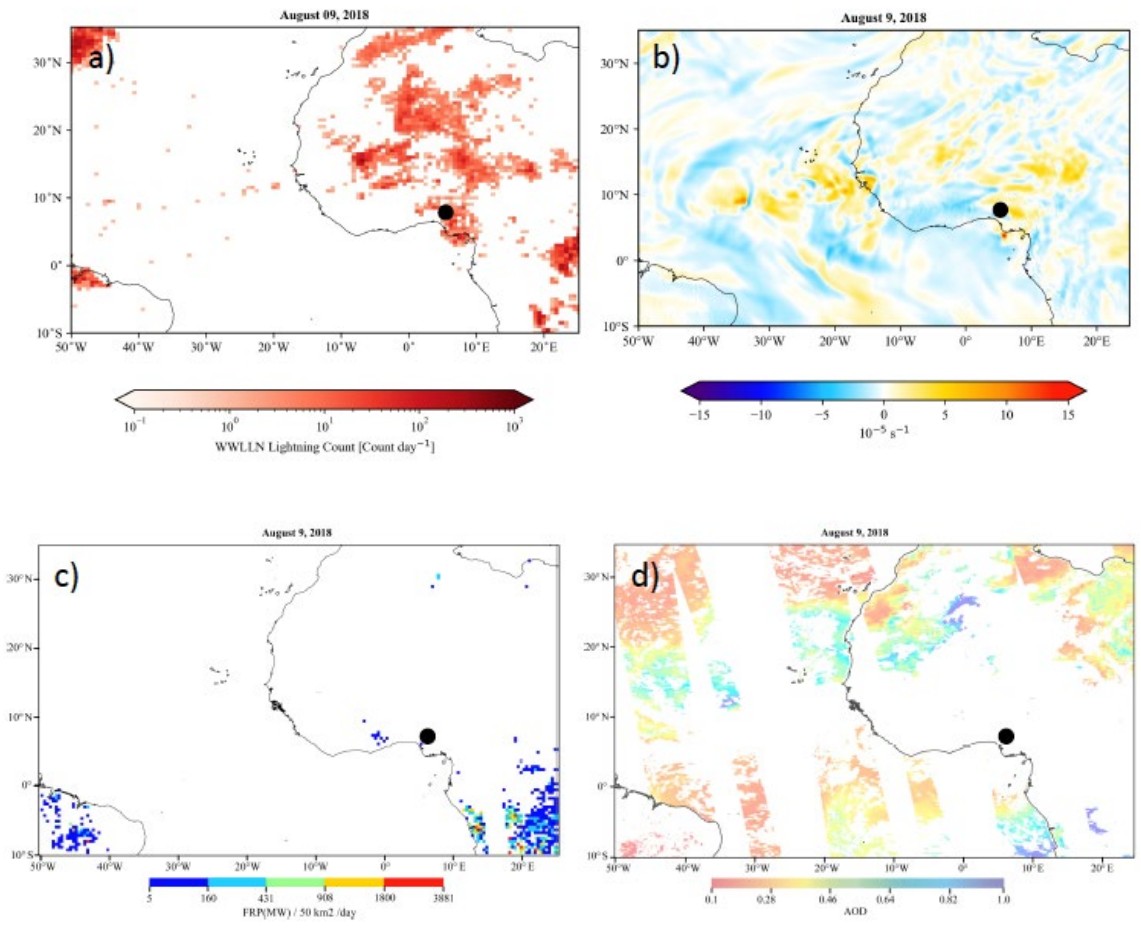

1025          *Figure 16. Same as Fig. 13 but for the EIE event recorded on 9 August 2018 (black dot).*

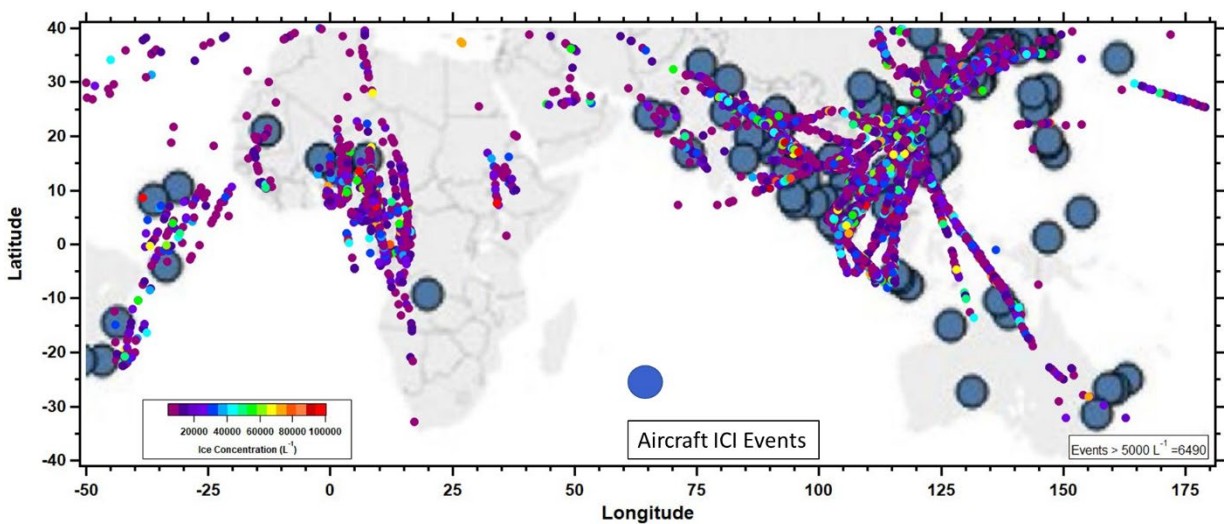

*Figure 17. EIE, color-coded by concentration, superimposed on the map of ice crystal icing (ICI) events (filled blue circles). The ICI map is reprinted from Bravin and Strapp (2020) with permission from the Society of Automotive Engineers (SAE) International Journal on Advances & Current Practices in Mobility.*


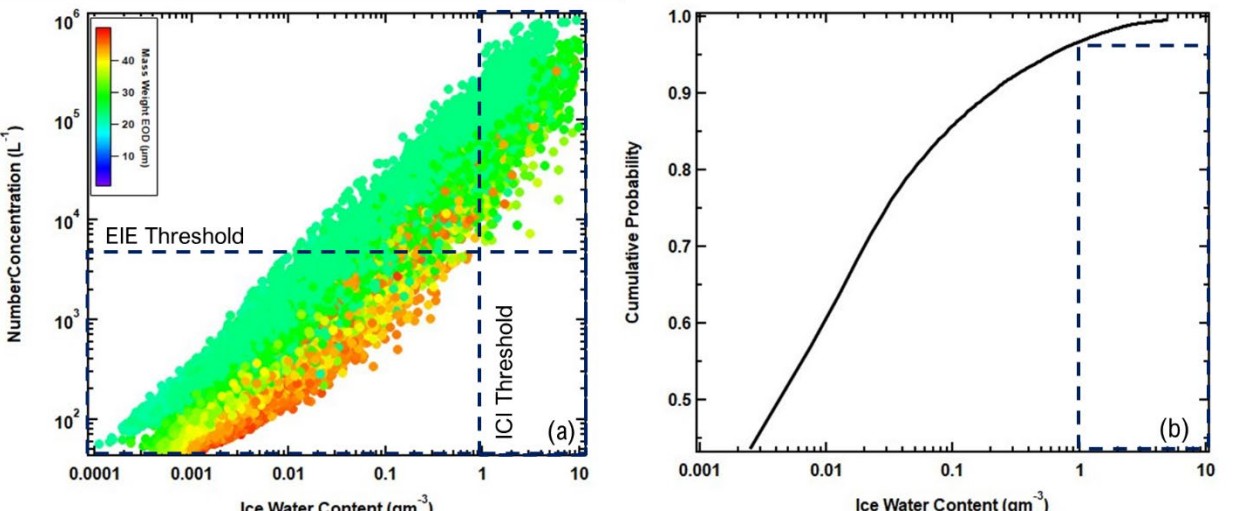
