# Peer review of "High Concentrations of Ice Crystals in Upper Tropospheric Tropical Clouds: Is there a Link to Biomass and Fossil Fuel Combustion?"

_Atmospheric Chemistry and Physics, 2021_

## Referee Comment (RC2)

In this paper the authors hypothesize that aerosols produced by biomass burning and urban pollution contribute to the occurrence of extreme ice crystal events (EIE). They assemble data sets from a diverse array of independent sources and demonstrate correlation between elevated aerosol amounts and clouds with ice number concentrations greater than 5000 L$^{-1}$. The paper is logically organized and clearly written. Analysis methods appear to be robust. The topic has significance for climate change research, weather forecasting, and aviation safety. I recommend that the paper be published with revisions as suggested below.

General Comments

The authors have shown qualitative correlation between several independent data sources and the IAGOS incidence of Extreme Ice Events. But their suggestion that high aerosol concentrations are a cause of EIE is still circumstantial. It would be interesting to somehow quantify typical values of AOD, CO anomalies, FRP and other indicators for the EIE vs. non-EIE samples, and to test the statistical significance of the differences between the two sample populations. Short of this, I believe the authors can clarify and streamline their arguments to present a more convincing case. At the same time, they should acknowledge more prominently that convective processes play a large role in determining the location of high ice water content regions.

Specific Comments

1. Introduction
   a. The beginning of the paper seems abrupt as the authors immediately reference prior publications by other researchers rather than introducing and motivating the topic of this paper. An introductory paragraph with some general information describing ice clouds, what is known about their interactions with aerosols, and/or why it matters would set-up the current study more effectively before reviewing past literature.
   b. Lines 75-79. Could the authors please explain why they chose to use $N_{ice}$ and define a new term (Extreme Ice Events) rather than using IWC? Some of the papers cited give existing thresholds for elevated ice conditions based on IWC (i.e., HIWC), so why not be consistent? If there is a valid reason for defining an EIE threshold rather than using existing HIWC thresholds, how do the two thresholds compare?
   c. Line 109. Baumgardner et al. (2004) is cited here, but not listed in the Reference section.
   d. Figure 1. Some of the black stars indicating megacities are obscured by the EIE symbols.

2. Measurement and Analysis Methodology
   a. Line 204 describes the geographic domain of the study as extending from -50$^{o}$ to +180$^{o}$ longitude, though Figure 1 indicates EIE outside of these boundaries and many of the subsequent figures (e.g., AOD, FRP) extend to a larger range of longitudes. It's not clear why the authors have limited their analysis to this region, or for that matter, why they didn't include events outside of the ±30$^{o}$ latitude range. Some explanation is in order.

3. Results
   a. Figures 6 and associated discussion. Do the authors have an explanation for why the measurement-derived median CO anomalies are ~5 times larger than the modeled CO anomalies? Also, I'm confused by the assertion that, "The frequency distributions do

suggest that emissions from UP sources are potentially a larger source of nucleating particles in the ice clouds, in general." Can you elaborate on how Figure 6 demonstrates this result?

4. Discussion
    a. Figure 10. The density of EIE events in SE Asia obscures the values of FRP in that region.
    b. Lines 364-66. The following statement suggests that the evidence presented thus far proves that aerosol particles are responsible for EIE: "However, the maps suggest that these BB emissions are the source of some, but not all of the particles that lead to EIE." While the authors have shown spatial and seasonal correlation between aerosol presence and ice crystal concentration, I think it is overstating to say that the particles cause EIE. Correlation is not causation, and as the authors discuss in later sections, other processes contribute to EIE.
    c. Line 372-373. Please explain why only certain aerosols are considered relevant. Would sea salt not be of interest, for example.
    d. Lines 378-380. The authors state that Figures 11 and 12 show the high AOD over northern Africa in July is collocated with high dust concentrations and EIE in this area. Maybe I'm misreading the small map in Figure 12, but the high dust concentration and cluster of EIE points appears to coincide with a relative minimum in the AOD distribution for July (top of Figure 11). Can the authors please clarify?
    e. Section 4.4. The authors nicely link their work to previous studies in this subsection. A key point in their argument is that the ice clouds observed in the IAGOS data set are likely liquid in origin. I'm confused about how the authors know this, and how it relates to their statement (in the abstract) that droplets are lofted and freeze heterogeneously. Please clarify.

5. Conclusions
    a. Lines 505-506. Regarding the qualitative comparison of ICI events and EIE, wouldn't a positive correlation be expected given that both phenomena are based on high amounts of ice crystals? This again raises my earlier question about why new terminology and associated threshold for high amounts of ice crystals is introduced for this analysis.

---

## Author Comment (AC1)

Response to All Reviewers

We are very grateful to the three reviewers, who have done an admirable job of critiquing our manuscript and raising important points related to the clarity of how we argue that there is a link between extreme ice concentrations at commercial flight altitudes and anthropogenic emissions at the surface. Many of the reviewers' comments which challenged our argument, have now been addressed. We have revised the Introduction that explains not only what the objectives of our study are, but that also emphasizes the uniqueness of this data set developed over nine years in what can be considered a totally random cloud sampling by commercial aircraft.

The reviewers criticize the study as having no quantitative evidence that supports our arguments for a link between surface sources and high ice concentrations. The only truly quantitative method to prove that such a link exists would be a Lagrangian study that measures aerosol properties at the surface and then follows these same particles as they form cloud hydrometeors. Given the near impossibility of such a study, we think that the methodology that we use, which combines quantitative aircraft measurements with multiple, independent data sets from satellite, a back-trajectory model and reanalysis, is as close to a quantitative evaluation as possible.

After reading the reviewers' comments, it became clear that we needed to state from the outset that our case was being built, by necessity, on circumstantial evidence but that the methodology that couples in situ and satellite measurements with atmospheric models makes a compelling argument for biomass burning and urban pollution as the most likely sources for the extreme ice events in the tropical regions evaluated.

We have made a number of modifications to the paper that we think will address many of the reviewers' concerns, and in particular, that we have been too aggressive in our conclusions regarding a causal link between high ice and anthropogenic emissions:

1. We have changed the title of the paper to "High Concentrations of Ice Crystals in Upper Tropospheric Tropical Clouds: Is there a Link to Biomass and Fossil Fuel Combustion?". This better represents our objectives while somewhat softening our conclusions.
2. The introduction has been rewritten to bring into sharper focus the objectives of the study, the uniqueness of the measurement platform and size of the data set, and to lay the foundation for our arguments that much of the high ice in the tropics is a result of ice-forming particles whose sources are anthropogenic emissions. The introduction now also mentions the two origins of upper troposphere ice clouds and why high ice concentration in such clouds are most likely of liquid origin, and how we connect surface sources to high ice concentrations using all the available resources at hand.
3. We have added a new subsection in the Discussion to highlight individual case studies to complement the larger data set from which our general conclusions are drawn. These case studies provide more direct evidence for the co-location of the

measured ice crystal concentrations, the surface sources and properties of potential cloud condensation nuclei (CCN) and Ice Nucleating Particles (INP) and the vertical transport mechanisms.

4. All reviewers have made comments that we have *repeatedly* asserted that high aerosol concentrations are the *cause* of high ice crystal concentrations. This was never our intent and as we look through the original manuscript, we only see a couple of times that we associate high ice with high aerosol concentrations. We have now removed those statements. In the new Introduction we explain that both biomass burning (BB) and urban pollution (UP) emissions are large area sources of particles, that the composition of many of these particles make them potential CCN or INP, and hence are a logical place to start investigating if there is a link with high ice crystal concentrations.

The point-by-point responses to reviewers are found below where we have listed each of the reviewers' comments, questions or recommendations followed by our responses highlighted in blue italics.

Response to Reviewer 1

1) I sense that the paper began as an investigation of aerosol effects on extreme ice events, as evidenced by the title, and also that it began with the notion that biomass and fossil fuel combustion particles were influencing these, but that the data do not back this up in more than a qualitative sense at best.

*We do not agree with this comment, and as explained in our opening comments above, we have used several, independent data sets and modeling, coupled with measurements, which support our conjecture of linking particles generated during combustion of biomass and fossil fuels to extreme ice.*

2) Further, it only becomes evident later in the paper (obvious to the reader, and then finally stated) that there is really no distinguishing factor between EIE and non-EIE events from the aerosol standpoint, and what is most driving EIE is deep convection.

*We partially agree that there seems to be no distinguishing factor between the EIE and non-EIE from the aerosol standpoint, at least not from the evidence that we are able to gather at this time. Whereas the reviewer might be correct that it is deep convection that leads to the higher concentrations, we cannot rule out the possibility that anthropogenic emissions are producing higher particle concentrations which lead to higher cloud droplet concentrations and, hence, would lead to higher ice crystal concentrations from droplet freezing in the updraft.*

3) Lots of strong updrafts lofting aerosols to low temperatures, potentially as liquid through homogeneous freezing conditions. Do the base aerosols even matter?

*Base aerosols matter since CCN or IN are needed to activate as cloud hydrometeors, and this is determined by the composition and their size. While stronger updrafts mean higher supersaturations and hence, activation of smaller CCN, the right composition and size distribution of aerosols are needed to activate. Many studies have shown, from measurements and cloud models, that higher CCN concentrations lead to higher cloud droplet concentrations for the same updraft velocity.*

4) The relation with aerosols is interesting but never proven to be causal. The remarkable consistency between regions and seasons in Fig. 2 does not to my mind speak to an obvious aerosol effect, and especially not a link to major cities.

   *There does not appear to be an actionable issue that we can respond to in this comment because in our discussion of Fig. 2, we make no mention of an aerosol effect or a link to major cities.*

5) Hence, one wonders about whether plots of CO or delta-CO versus ice concentrations are not shown. All else being equal, would a relationship not be expected?

   *As we have now included in the introduction, and also reiterate in the discussion and conclusions, the magnitude of CO concentrations, as well as the CO anomalies, cannot be used directly as a proxy for either the magnitude of the ice crystal concentrations or precursor aerosols. The reason is because different processes remove or modify aerosols and CO concentrations. The CO is only being used as a tracer of the air masses that bring CO and aerosol particles from surface emissions to flight level. There was no physical basis for trying to correlate CO anomalies with ice concentrations*

6) And what would be expected if the aerosol effect were related to either heterogeneous versus homogeneous freezing nucleation?

   *Although this question is somewhat vague as to its intent, we believe that we have now better clarified in the introduction, and reinforced in the summary, that the clouds that we are studying are for the most part form by frozen water droplets and the freezing mechanism probably has little relevance, i.e., whether the droplets freeze homogeneously at temperatures colder than -38°C or heterogeneously at warmer temperature. In the presence of the strong updrafts that loft these frozen droplets to the UTLS the only process that matters is the formation and evolution of the water droplets.*

7) Would dust influences be distinguished from the others?

   *From a cloud microphysical perspective, yes, since dust tends to be better INP than CCN; however, studies have also shown that even with a little aging that can deposit hygroscopic material on their surface, dust particles can be good CCN. Hence, although we can't say unequivocally that dust would be indistinguishable from other*

*aerosols, without having a counterflow virtual impactor (CVI) to extract the ice crystal residuals, we are unable to distinguish dust from our analysis of the measurements. As the reviewer is aware, as reflected in the references provided at the end of his/her review, there are a very limited number of cirrus studies using a CVI on the WB-57 and many of the residuals were identified as dust particles.*

8) The anomaly plots are not all especially revealing about what is driving things.

   *We agree, and have removed Fig. 6, for the reasons that we put forward above and have now put into the introduction regarding the lack of correlation between CO and ice crystal concentration magnitudes. We have also modified the text throughout to remove any discussion that would even imply that such a correlation should exist.*

9) Describe in the introduction how aerosols might influence cirrus concentrations via heterogeneous AND homogeneous freezing.

   *A discussion to this effect is now included in the Introduction.*

10) Discuss the complicating role of cloud dynamics and convection, and how the nature of the cirrus targeted (if all expected to be liquid-formed) matters. This and the first suggestion add context to the study, instead of diving immediately into aerosols as the only influence.

    *We have made clear in the revised Introduction that vertical motion and boundary layer aerosols are two essential components in the clouds that were sampled.*

11) Consider if some truly quantitative analyses of relations between aerosols/gases and ice concentrations are possible, instead of only associations of fires, smoke and pollution areas with areas of EIE.

    *As we explain in our opening comments before the point-wise responses, we think that the use of multiple, independent sources of data is as close to a quantitative assessment as possible without a Lagrangian measurement program, which has not been done for this type of clouds, to the best of our knowledge.*

12) Abstract: Line 25: Using the term ice-forming aerosols is not exact, in that the aerosols may or may not be directly linked to the freezing mechanism. If homogeneous freezing, the factor of importance is simply that the particles carry liquid with them. If heterogeneous freezing, the nature of the particles truly matters. Perhaps, lofting aerosols that directly or indirectly lead to freezing, or better serve as seeds for heterogeneous and homogeneous freezing nucleation?

    *While we understand the reviewer's point, although the term we use may not be exact, neither is it incorrect and we are using it as a short form description without having to add qualifiers. Nevertheless, we do use qualifiers later in the Introduction to explain the role of aerosols in the formation of ice crystals.*

13) Lines 28-29: Why only heterogeneously if the cold clouds are of liquid origin? There would be a competition between heterogeneous freezing and homogeneous freezing, and what wins at cloud top will be determined by both cloud dynamics (updraft) supplying supersaturation and the propensity of particles for freezing heterogeneously and growing prior to the point where homogeneous freezing will ensue. Where would 5000 per liter INPs come from prior to -38°C? And what concentrations would be necessary freezing prior to that temperature to defeat water persistence to -38C that could then lead to further massive freezing? Consider the observations of Rosenfeld and Woodley (2000) in this regard. Deep convective clouds readily overcome the relatively low numbers of INP from the boundary layer.

*We removed the word "heterogeneously" and have expanded the discussion of ice cloud origins in the Introduction.*

14) Introduction: Lines 81-84 paragraph: In reference to the point above, homogeneous freezing needs mention as a potentially very important process.

*We have expanded the discussion of ice cloud origins in the Introduction and added homogeneous freezing as one of the possible processes that produce ice crystals.*

15) Lines 90-92: "Some fraction of particles emitted from biomass and fossil fuel burning will act as CCN or INP especially as they age while lofted to the UT…". I consider the especially while they age part as not yet strongly demonstrated for the ambient atmosphere. Atmospheric measurements in this regard are not well-represented in the reference list. Recently, both Schill et al. (2020) and Barry et al. (2021) discuss ambient measurements related to biomass burning INPs, and production is mentioned in the latter study. Those measurements directly in plumes should constrain expectations on INP concentrations feasible from biomass burning, at least at temperatures in the mixed-phase regime prior to the homogeneous freezing threshold.

*We agree with the reviewer and have removed "especially as they age" and added references to the Barry et al. and Schill et al. studies.*

16) introduction and Figure 1: EIEs appear to occur in all regions, and quite high values occur even over oceans. Realizing that your focus is on connecting certain sources and EIEs, I wondered if the IAGOS network coverage adds any particular bias. Does the absence of occurrences between Japan and the U.S. indicate a true absence or a limitation of the network? In this regard, I felt it would be helpful to see a supplemental figure of all of the flight paths. Then it would be easier to understand where flying occurred versus where high values were seen. This point about potential bias or absence of coverage is only otherwise brought up late in the paper on lines 412-413, in regard to absence of flights over a region in Africa. Yet, large ocean regions of the Pacific in both hemispheres are missing from assessment, in a

region more remote from fire and urban influences. As a second comment, some of the data are from over the maritime continent and other open ocean regions. Is there a reason not to consider sea salt as an aerosol that could affect deep convective clouds? It has been noted as a freezing nucleus at low temperatures in laboratory studies (e.g., Wagner et al., 2018), and was identified in ice residuals in deep convective anvils (Cziczo et al., 2013). When one is dealing with aerosols and ice nucleation, abundance and activation potential are both factors to consider, so I do not see a reason to exclude something in favor of something more abundant like biomass burning particles. This also arises later when the aerosols of "relevance" are mentioned on lines 373-374. It is simply what you chose to focus on.

*The reviewer is correct that there are large regions that unfortunately remain uncovered by the current IAGOS network. As suggested, we are adding the flight tracks to the supplemental material and in the introduction we point out the data gaps.. As we have now explained in the revised text, there are two reasons that sea salt is not included as a significant contributor to EIE: 1) sea salt aerosols, while excellent CCN, are also quite large compared to CCN from BB or UP emissions. Hence, these will be the first to form large water droplets that will very rapidly grow to precipitation-sized drops and be removed as rain before ever reaching the UTLS and 2) The MERRA results showed mass concentrations of sea salt at flight altitudes that were significantly lower than the sulfate or OC/BC at those altitude where EIE were found.*

17)  Results: Section 3.3 and Figure 6: I feel that a better explanation of the meaning of this figure is needed. What is event frequency? Is it any concentrations of ice crystals coinciding with a CO anomaly? I see nothing much distinguishing low-ice and extreme-ice, and the values of the median CO anomalies are extraordinarily low compared to say CO anomalies inside and outside of biomass burning plumes. How does this indicate impact, if at all? Or especially, how does it show that "…frequency distributions do suggest that emissions from UP sources are potentially a larger source of nucleating particles in the ice clouds, in general."? To me, I interpret this figure to mean that clouds and aerosols will be associated, but there is no smoking gun for any particular aerosol type or its direct involvement in creating EIEs.

We have removed Figure 6 and modified the discussion here and throughout the manuscript, emphasizing that the anomalies are only being used to identify source regions with no intent to suggest that their magnitudes are necessarily proportional either to the intensity of the emissions or the concentration of the ice. Because the model does not specifically provide a time between when the CO was found in the clouds and when it left the surface, the magnitude of the CO anomaly may be impacted by dilution along the trajectory and should not be correlated with source intensity or particle concentration.

18) Discussion: Lines 320-321: This inclusion of CCN here may be a nod to homogeneous freezing as a source of ice clouds, but only the INP connection is tendered earlier as a hypothesis. If the different mechanisms are made explicit in the introduction, this will all be resolved.

*The Introduction now resolves this issue.*

19) Line 345: AOD is an integrated measure. You do not know where in the vertical it resides, right? Most often it is in the boundary layer, although I understand that plumes can be elevated. And it seems that the full range of AOD underlies the EIE points. Like fire power and other relations, the correlation is only a spatial one as viewed from above.

*Yes, the AOD is an integrated measure and we are using monthly AODs averaged over the same nine years as the cloud data as an additional independent data set. We are showing regions that climatologically have high aerosol loading. We have now added case studies to provide a more direct connection between the emission sources at the surface, high AOD and deep convection.*

20) Line 375: Fig. SMx? There is no AOD plot of this type in the supplement.

*Supplemental figure 2 (SM2) shows all 12 months of AOD. Apologies, SMx was supposed to be SM6, the MERRA maps of particle mass for the dust, sulfate and OC+BC*

21) Line 393, paragraph: Not much is said about AOD over parts of Indonesia to Australia, which are striking for the apparent lack of any apparent influence. This is the regions that begs explanation, if it is to be contended that only certain types of particles are associated with EIEs.

*The reviewer raises an important omission that we have corrected in the revised manuscript. The lack of high AODs over the Indonesia is due to the generalized presence of clouds since cloud filtering is necessary to determine AOD. Looking at the OLR and lightning plots, which are a measure of cloud cover and convective activity, the reviewer will see that the region where there is no AOD data corresponds to moderate-to-high OLR and extensive lightning associated with deep convection.*

22) Section 4.2: Not intending to beat on a point I raised already in summary, but convection so clearly shows the strongest correlation with EIE, regardless of aerosols. One has to ask for more than association with CO and other tracers in order to claim that any specific aerosol type is making a difference. Regardless, I felt that the convection link came far too late in this paper, and has to raise a question about the appropriateness of the title.

*With the modification of the title, the expanded Introduction, additional discussion related to lightning and upper-level divergence maps and circling back to convection*

*in the summary and conclusion, we think that we have adequately addressed the reviewer's concern that we don't give convection enough credit for the EIE.*

23) Section 4.4: A similar comment about a summary point. The discussion under this section finally acknowledges the ways that aerosols, ice formation and deep convection interplay, including mention of liquid origin ice processes and homogeneous freezing. Missing still is the fact that there must also be a relation between ice concentration and vertical velocity. And the mechanism will depend on that and on the freezing efficiency of heterogeneous INPs, as mentioned earlier. This should have been discussed up front, rather than alluding to the fact that the mechanism might be via INPs only.

*We think that we have now addressed this in multiple places in the manuscript as has been highlighted in our responses above.*

24) Line 386: Intended reference is missing.

*We have added not only the Demott reference but also the Czizco reference that the reviewer provided. In addition, we would like to express our appreciation to the reviewer for all the references that were provided.*

25) Line 485: "temperature of EIE.."

*We have corrected this typo.*

---

## Author Comment (AC2)

Response to All Reviewers

We are very grateful to the three reviewers, who have done an admirable job of critiquing our manuscript and raising important points related to the clarity of how we argue that there is a link between extreme ice concentrations at commercial flight altitudes and anthropogenic emissions at the surface. Many of the reviewers' comments which challenged our argument, have now been addressed. We have revised the Introduction that explains not only what the objectives of our study are, but that also emphasizes the uniqueness of this data set developed over nine years in what can be considered a totally random cloud sampling by commercial aircraft.

The reviewers criticize the study as having no quantitative evidence that supports our arguments for a link between surface sources and high ice concentrations. The only truly quantitative method to prove that such a link exists would be a Lagrangian study that measures aerosol properties at the surface and then follows these same particles as they form cloud hydrometeors. Given the near impossibility of such a study, we think that the methodology that we use, which combines quantitative aircraft measurements with multiple, independent data sets from satellite, a back-trajectory model and reanalysis, is as close to a quantitative evaluation as possible.

After reading the reviewers' comments, it became clear that we needed to state from the outset that our case was being built, by necessity, on circumstantial evidence but that the methodology that couples in situ and satellite measurements with atmospheric models makes a compelling argument for biomass burning and urban pollution as the most likely sources for the extreme ice events in the tropical regions evaluated.

We have made a number of modifications to the paper that we think will address many of the reviewers' concerns, and in particular, that we have been too aggressive in our conclusions regarding a causal link between high ice and anthropogenic emissions:

1. We have changed the title of the paper to "High Concentrations of Ice Crystals in Upper Tropospheric Tropical Clouds: Is there a Link to Biomass and Fossil Fuel Combustion?". This better represents our objectives while somewhat softening our conclusions.
2. The introduction has been rewritten to bring into sharper focus the objectives of the study, the uniqueness of the measurement platform and size of the data set, and to lay the foundation for our arguments that much of the high ice in the tropics is a result of ice-forming particles whose sources are anthropogenic emissions. The introduction now also mentions the two origins of upper troposphere ice clouds and why high ice concentration in such clouds are most likely of liquid origin, and how we connect surface sources to high ice concentrations using all the available resources at hand.
3. We have added a new subsection in the Discussion to highlight individual case studies to complement the larger data set from which our general conclusions are drawn. These case studies provide more direct evidence for the co-location of the

measured ice crystal concentrations, the surface sources and properties of potential cloud condensation nuclei (CCN) and Ice Nucleating Particles (INP) and the vertical transport mechanisms.

4. All reviewers have made comments that we have *repeatedly* asserted that high aerosol concentrations are the *cause* of high ice crystal concentrations. This was never our intent and as we look through the original manuscript, we only see a couple of times that we associate high ice with high aerosol concentrations. We have now removed those statements. In the new Introduction we explain that both biomass burning (BB) and urban pollution (UP) emissions are large area sources of particles, that the composition of many of these particles make them potential CCN or INP, and hence are a logical place to start investigating if there is a link with high ice crystal concentrations.

The point-by-point responses to reviewers are found below where we have listed each of the reviewers' comments, questions or recommendations followed by our responses highlighted in blue italics.

Response to Reviewer 2

1) The authors have shown qualitative correlation between several independent data sources and the IAGOS incidence of Extreme Ice Events. But their suggestion that high aerosol concentrations are a cause of EIE is still circumstantial.

*As we emphasize in our opening remarks to the reviewers, and have now clarified in the revised Introduction, we did not intend to assign causality between high aerosol concentrations and EIE. Any such references to such a link have now been removed.*

2) It would be interesting to somehow quantify typical values of AOD, CO anomalies, FRP and other indicators for the EIE vs. non-EIE samples, and to test the statistical significance of the differences between the two sample population.

*The case studies that we have added to the manuscript partially address the reviewer's interest in quantifying the indicators of anthropogenic emissions at the surface with the EIE. However, please note that we do not have in-situ measurements, and are only using satellite and reanalysis data sets. We concur that rigorous statistical testing is more satisfying when accepting or rejecting a hypothesis; however, as we discuss in our opening comments to the reviewers, it is the nature of the problem that makes such statistical testing impossible without Lagrangian measurements.*

3) Introduction a. The beginning of the paper seems abrupt as the authors immediately reference prior publications by other researchers rather than introducing and motivating the topic of this paper. An introductory paragraph with some general information describing ice clouds, what is known about their interactions with

aerosols, and/or why it matters would set-up the current study more effectively before reviewing past literature.

*The Introduction has been substantially modified so that now we set the stage for the remainder of the paper by discussing the two primary ways that ice clouds in the UTLS are formed, i.e. in situ and liquid origin, and how the liquid origin pathway is the most likely for the clouds we analyzed. We also discuss the importance of vertical motion along with the aerosol type and use these discussions to then explain why we are focusing on identifying the source of the aerosols on which ice crystals form in the EIE.*

4)  Lines 75-79. Could the authors please explain why they chose to use Nice and define a new term (Extreme Ice Events) rather than using IWC? Some of the papers cited give existing thresholds for elevated ice conditions based on IWC (i.e., HIWC), so why not be consistent? If there is a valid reason for defining an EIE threshold rather than using existing HIWC thresholds, how do the two thresholds compare?

*In the revised Introduction, we clarify why we prefer to use Extreme Ice Events that use $N_{ice}$ as the metric rather than IWC or HIWC. Research projects that investigate HIWC use thermal devices that measure IWC directly with less uncertainty than the method used in our study. We derive IWC from the BCP-measured size distribution, and estimate that the derived IWC has a >±50% uncertainty. In contrast, $N_{ice}$, can be determined with an accuracy of ±15% because it doesn't depend on any assumption about ice density or particle shape or size, which gives us more confidence on our results. Nevertheless, we have now added a new figure (see below) that shows the IWC vs $N_{ice}$ in the sub-section on aircraft operations impact in the Discussion (included below). We include this to show that our derived IWC is above 1 $gm^{-3}$ in more than 2000 of the EIE clouds, i.e. meeting the threshold of the HAIC community.*

[Figure]

.

5) Line 109. Baumgardner et al. (2004) is cited here, but not listed in the Reference section.

*This reference has now been added.*

6) Figure 1. Some of the black stars indicating megacities are obscured by the EIE symbols.

*We have redrawn the maps to bring the megacity stars to the front.*

[Figure]

7) Measurements: Line 204 describes the geographic domain of the study as extending from -50º to +180º longitude, though Figure 1 indicates EIE outside of these boundaries and many of the subsequent figures (e.g., AOD, FRP) extend to a larger range of longitudes. It's not clear why the authors have limited their analysis to this region, or for that matter, why they didn't include events outside of the ±30º latitude range. Some explanation is in order.

*The revised Introduction explains the reason that we have focused on low latitudes and Figure 1 (see figure shown in previous response) is now constrained to ±30º. In short, the highest density of EIE are in the tropics, with the largest regions of anthropogenic biomass burning and several rapidly-growing megacities.*

8) Results: Figures 6 and associated discussion. Do the authors have an explanation for why the measurement-derived median CO anomalies are ~5 times larger than the modeled CO anomalies? Also, I'm confused by the assertion that, "The frequency distributions do suggest that emissions from UP sources are potentially a larger source of nucleating particles in the ice clouds, in general." Can you elaborate on how Figure 6 demonstrates this result?

*We have modified the text and removed Fig. 6 to clarify our message that we are not using the CO anomaly as a proxy for either the ice crystal concentration or intensity of the combustion emissions. In our original analysis we had evaluated correlations between CO anomalies and ice crystal concentrations but concluded that no such correlation should be expected because CO and aerosol particles are removed or modified by almost completely independent processes. Hence, the CO anomaly is only a way to identify general aerosol source areas, not aerosol concentrations. Throughout the manuscript we try to return to the basic argument that the CO anomaly identifies elevated air masses under the influence of emissions larger than the background, the back trajectory analysis is used to identify from which region this air came from and if the CO is a result of BB or UP. The remaining piece of the puzzle is to identify the mechanism that transports this air to flight levels. This we have done with the best tool available, i.e. the reanalysis of the meteorological variables that provide the low-level convergence and upper-level divergence fields associated convection in reanalysis, as well as outgoing longwave radiation (OLR) to indicate the presence of clouds and lightning as proxy for deep convection.*

9) Discussion: Figure 10. The density of EIE events in SE Asia obscures the values of FRP in that region.

*Although we understand the reviewer's comment, we can't see an easy way to show both the important EIE and FRP so we have chosen to keep the EIE on top.*

10) Lines 364-66. The following statement suggests that the evidence presented thus far proves that aerosol particles are responsible for EIE: "However, the maps suggest that these BB emissions are the source of some, but not all of the particles

that lead to EIE." While the authors have shown spatial and seasonal correlation between aerosol presence and ice crystal concentration, I think it is overstating to say that the particles cause EIE. Correlation is not causation, and as the authors discuss in later sections, other processes contribute to EIE.

*The case studies in the revised manuscript provide more evidence, but we accept that the reviewer considers it an overstatement and have now modified this text.*

11) Line 372-373. Please explain why only certain aerosols are considered relevant. Would sea salt not be of interest, for example.

*As we have now explained in the revised text, there are two reasons that sea salt is not included as a significant contributor to EIE: 1) sea salt aerosols, while excellent CCN, are also quite large compared to CCN from BB or UP emissions. Hence, these will be the first to form large water droplets that will very rapidly grow to precipitation-sized water drops and be removed as rain before ever reaching the UTLS and 2) The MERRA results showed mass concentrations of sea salt at flight altitudes that were significantly lower than the sulfate or OC/BC at those altitude where EIE were found.*

12) Lines 378-380. The authors state that Figures 11 and 12 show the high AOD over northern Africa in July is collocated with high dust concentrations and EIE in this area. Maybe I'm misreading the small map in Figure 12, but the high dust concentration and cluster of EIE points appears to coincide with a relative minimum in the AOD distribution for July (top of Figure 11). Can the authors please clarify?

*As we now expand upon in the revised manuscript, AOD cannot be retrieved in the presence of generalized clouds. So what this means is that since the AOD maps are averages over nine years the very high AOD regions just north of the majority are indeed associated with the MERRA maps of dust in Fig. 12 but the region where the EIE are is also the region where there is a lot of deep convection during the July time period as is shown in Figure 13 where the region of lower AOD is where the 200 mb divergence is high, indicating strong upward motion, the OLR is a minimum, because the clouds are blocking the outgoing radiation, and lightning activity is a maximum, again associated with a lot of deep convection and ice. We have now added discussion to the text that clarifies this point.*

13) Section 4.4. The authors nicely link their work to previous studies in this subsection. A key point in their argument is that the ice clouds observed in the IAGOS data set are likely liquid in origin. I'm confused about how the authors know this, and how it relates to their statement (in the abstract) that droplets are lofted and freeze heterogeneously. Please clarify.

*As suggested by all the reviewers, the liquid origin assumption is discussed in the Introduction as described by Krämer et al (2016) who explain why the tropics, with*

*their deep convection and thicker clouds are the most likely to form by the liquid origin mechanism.*

14) Conclusions: Lines 505-506. Regarding the qualitative comparison of ICI events and EIE, wouldn't a positive correlation be expected given that both phenomena are based on high amounts of ice crystals? This again raises my earlier question about why new terminology and associated threshold for high amounts of ice crystals is introduced for this analysis.

*We explain in the revised paper that ICI is used by the community that studies the impacts of high crystal concentrations on aircraft performance. The goal of our study is to contribute to improving our understanding of the processes that lead to the presence of high ice concentrations in upper-troposphere clouds analysing the IAGOS database.*

---

## Author Comment (AC3)

**Response to Reviewers**

We are very grateful to the three reviewers, who have done an admirable job of critiquing our manuscript and raising important points related to the clarity of how we argue that there is a link between extreme ice concentrations at commercial flight altitudes and anthropogenic emissions at the surface. Many of the reviewers' comments which challenged our argument, have now been addressed. We have revised the Introduction that explains not only what the objectives of our study are, but that also emphasizes the uniqueness of this data set developed over nine years in what can be considered a totally random cloud sampling by commercial aircraft.

The reviewers criticize the study as having no quantitative evidence that supports our arguments for a link between surface sources and high ice concentrations. The only truly quantitative method to prove that such a link exists would be a Lagrangian study that measures aerosol properties at the surface and then follows these same particles as they form cloud hydrometeors. Given the near impossibility of such a study, we think that the methodology that we use, which combines quantitative aircraft measurements with multiple, independent data sets from satellite, a back-trajectory model and reanalysis, is as close to a quantitative evaluation as possible.

After reading the reviewers' comments, it became clear that we needed to state from the outset that our case was being built, by necessity, on circumstantial evidence but that the methodology that couples in situ and satellite measurements with atmospheric models makes a compelling argument for biomass burning and urban pollution as the most likely sources for the extreme ice events in the tropical regions evaluated.

We have made a number of modifications to the paper that we think will address many of the reviewers' concerns, and in particular, that we have been too aggressive in our conclusions regarding a causal link between high ice and anthropogenic emissions:

1. We have changed the title of the paper to "High Concentrations of Ice Crystals in Upper Tropospheric Tropical Clouds: Is there a Link to Biomass and Fossil Fuel Combustion?". This better represents our objectives while somewhat softening our conclusions.
2. The introduction has been rewritten to bring into sharper focus the objectives of the study, the uniqueness of the measurement platform and size of the data set, and to lay the foundation for our arguments that much of the high ice in the tropics is a result of ice-forming particles whose sources are anthropogenic emissions. The introduction now also mentions the two origins of upper troposphere ice clouds and why high ice concentration in such clouds are most likely of liquid origin, and how we connect surface sources to high ice concentrations using all the available resources at hand.
3. We have added a new subsection in the Discussion to highlight individual case studies to complement the larger data set from which our general conclusions are drawn. These case studies provide more direct evidence for the co-location of the

measured ice crystal concentrations, the surface sources and properties of potential cloud condensation nuclei (CCN) and Ice Nucleating Particles (INP) and the vertical transport mechanisms.

4. All reviewers have made comments that we have *repeatedly* asserted that high aerosol concentrations are the *cause* of high ice crystal concentrations. This was never our intent and as we look through the original manuscript, we only see a couple of times that we associate high ice with high aerosol concentrations. We have now removed those statements. In the new Introduction we explain that both biomass burning (BB) and urban pollution (UP) emissions are large area sources of particles, that the composition of many of these particles make them potential CCN or INP, and hence are a logical place to start investigating if there is a link with high ice crystal concentrations.

The point-by-point responses to reviewers are found below where we have listed each of the reviewers' comments, questions or recommendations followed by our responses highlighted in blue italics.

Response to Reviewer 3

1) My most important concern is that almost all statements in the manuscript that attempt to link EIE to aerosol sources are NOT well grounded. Some examples are Lines 25-27, 291-292, 337-339, 356-358, 364-365, 381, 391, 396-397, 446-447, 450-451, 490. The authors repeatedly attribute the EIE to high aerosol concentrations nearby.

*The reviewer's assertion that our linkage of EIE to aerosol is not well-grounded is difficult to rebut without the reviewer offering a specific counter example of what would be considered a well-grounded argument. The reviewer also asserts that we "repeatedly attribute the EIE to high aerosol concentrations nearby". Below are listed, in quotations, all of the statements the reviewer lists as examples of our poorly-grounded arguments.*

*i)"The MERRA-2 analysis shows clear spatial correlations that link dust, black carbon (BC), organic carbon (OC) and sulfate particles with regions of EIE."*
*ii)"The frequency distributions do suggest that emissions from UP sources are potentially a larger source of nucleating particles in the ice clouds, in general."*
*iii)"In December there are EIE along the airline route between Northern Africa and South America, these would appear to be related to enhanced emissions of BB in Northern Africa and westward transport, as is discussed below."*
*iv)" Nevertheless, this region adjoins the area of most frequent EIE indicating high aerosol particle concentrations associated with the ice clouds."*
*v)" The proximity of the EIE to regions with large magnitude of AOD suggests that these clouds have likely formed on aerosol particles from relatively nearby sources"*
*vi)"This strongly suggests that the EIE in this region is likely related to dust, in addition to the BB that is also adjoining this region during July and whose presence is confirmed by the CO analysis."*

*vii)"Hence, the particles associated with the BB emissions are clearly linked with the fires, CO, OC/BC and EIE.*

*viii)"These high concentrations are partially reflected in the larger AOD, but the particularly striking feature are the many EIE in the region over eastern Asia.*

*ix)" The presence of ice clouds with extremely high crystal concentrations, clouds that in this current study have been associated with ground based emissions of anthropogenic CO and aerosol particles."*

*x)"The results that we have presented provide a framework for linking ice clouds in general, and EIE in particular, to surface sources of dust, BB and UP in tropical latitudes."*

*xi)" We conclude that the two, primary factors that are associated with the EIE encounters are the proximity to sources of dust, OC/BC or sulfate combined with strong vertical motions in deep convective clouds"*

*Note that of the 11 examples, only in iv) do we associate high concentrations of aerosol particle with high concentrations of ice crystal and we have now modified that text. Note, however, that the aerosol optical depth (AOD) is directly proportional to the vertical integral of the aerosol concentration and since the value of the AOD was much higher than surrounding areas, to attribute the high AOD to high aerosol concentrations was not totally incorrect.*

*We understand that the reviewer is not convinced by our evidence. We have now modified the text to carefully state only what the figures show. We have also included a few case studies on specific dates that provide more support for our statements. We have also toned down some of our conclusions to highlight the uncertainties involved.*

2)  However, according to Figure 6, the overall CO concentrations are even slightly lower in EIE as compared to the scenes with low ice concentrations

*As we now explain in the revised manuscript, we use the back trajectories of the CO to identify the most likely source of the air masses in which clouds form, and do not attribute higher aerosol concentrations to higher CO anomalies. We do assert that the aerosols on which cloud particles formed are from the same source as the CO. Since the processes that remove aerosols and CO from the air masses are different, we do not use CO anomalies as proxy for aerosol concentrations.*

3)  Besides, Figure 11 shows that, while some EIEs do occur in the vicinity of high AOD, even a larger number of EIEs occur in regions with quite low AOD.

*The revised text clarifies that the AOD is used to identify the source regions of those aerosols on which water droplets and ice crystals for, and not the regions of EIE. The reason for this, as the reviewer points out, is that regions of EIE are also regions of frequent clouds but low AOD. This is because in order to derive AOD measurements the algorithm removes data points where clouds have been*

*identified. Hence, the low AOD is not because there are no aerosols in these regions but because the frequent presence of clouds prevents the estimate of AOD. By comparing the maps of outgoing longwave radiation (OLR) and lightning in Fig. 12 (which indicate the presence of clouds and deep convection, respectively), it is evident that the regions of high EIE but low AOD, are regions where, on average over the 9 years of data, that there is extensive cloud activity. The new case studies discussing specific dates, show the absence of AOD data in the presence of high lightning activity.*

4) After reading the manuscript, my impression is that the current results can hardly support any causal relationship between the occurrence of EIE and the occurrence of high aerosol concentrations. Please carefully reevaluate all related statements throughout the manuscript and either remove them or provide convincing supporting evidence.

   *We have followed the reviewer's recommendation starting with the abstract and continuing through the summary and conclusions.*

5) Also, in view of the above comments, the last two objectives stated in Line 136-139 are not appropriate

   *The revised text now states: The four objectives of the study are: 1) to document the frequency of EIE by geographic region within the latitude band most impacted by BB and UP emissions, 2) to evaluate the seasonal variations of EIE as related to dry and rainy periods, 3) to identify regional sources of INP most closely associated with the EIE and 4) to show that there is sufficient convection to transport these INP, and the cloud particles that form on them into the UTLS.*

6) Line 20: not only anthropogenic sources but also biomass burning

   *Line 20 is now modified to read "Evaluation of in situ measurements of carbon monoxide in these UT clouds, combined with back-trajectories and carbon monoxide emission inventories, identified regions of potential, anthropogenic sources of ice crystal forming particles." We consider biomass burning to be anthropogenic at low latitudes, for the most part associated with land clearing for agricultural purposes and with burning of refuse after harvest. In other parts of the world, such as temperate forests, BB can be accidental or lightning induced, but not typically in tropical regions of South America, Asia or Africa.*

7) Line 168-171: Please provide more details about the SOFT-IO tool since most readers are probably not familiar with it. How does this tool link in situ detected CO to emission sources? What are the main inputs to the tool?

   *The revised text has been expanded to clarify the way SOFT-IO links the in situ Coat flight level to the emission sources and what the main inputs are to this back-trajectory analysis.*

8) Line 205: Please show the spatial extents of these four regions in at least one figure in the main text.

*Figure 1 now has the four regions outlined with dashed, colored lines, as shown here below.*

[Figure]

9) Line 240: This paragraph can be moved to the Method section.

*As the reviewer notes, this is repetitive as it has already been introduced in the methodology section so we just removed it but added a brief introduction of the results subsections.*

10) Line 386: Correct the typo here.

*Corrected, i.e., reference has been added.*

11) I suggest that the error bars be added to Figures 4 and 8.

*Added as suggested.*

12) Tables 1-2 can be moved to the Supplementary Information.

*Moved as recommended.*

---

## Author Response (AR2)

**A Note to All of the Reviewers**

The authors are appreciative of the detailed attention that the reviewers have paid to the reading and review of our manuscript. In particular, their level of understanding of cloud dynamics and microphysics have provided us with a perspective that is on occasions, somewhat different than ours. Their comments have reminded us that not all of our readers may have the same level of experience measuring cloud properties as we have, nor the familiarity of how clouds in the tropics differ from those that form in other latitudes. Hence, we have tried to be responsive to the opinions put forth by the reviewers while also maintaining our own interpretation of our measurement results. All three reviewers have acknowledged the value of this data set and, to our knowledge, no other study has taken the analysis of these cloud measurement to the depth that we have, especially assembling auxiliary satellite and reanalysis products to support our case for how ice clouds form in the UTLS in tropical latitudes.

Our observational study is, nevertheless, unable to produce the "smoking gun" that unequivocally links ice crystal microphysics to particle emissions at the surface. To do so will require modeling expertise beyond the scope of the current paper and by modelers more skilled at this activity than our team, whose expertise is in measurements and data interpretation. The reviewers have offered their own explanations of how these clouds might form and we have incorporated at least a summary; however, in our opinion, based upon all the evidence that we have assembled, our explanations are more likely and the reviewers have not offered evidence to the contrary.

Response to Referee #1

1) The last round of revision has improved the manuscript since the authors have removed or at least weakened many statements regarding the causal relationships between extreme ice crystal events (EIE) and emissions from biomass burning and urban pollution. However, there remain some statements that articulate or suggest the physical links between the two. After reading the revised manuscript and the authors' response to the reviewers' comments, I insist that the results presented in this manuscript cannot prove any physical links between EIE and biomass burning/urban pollution aerosols. Instead, the results only show that CO/aerosols from biomass burning/urban pollution often coincide or coexist with EIE. As one reviewer pointed out and the authors also agreed on, there is really no distinguishing factor between EIE and non-EIE events from the aerosol standpoint. For example, it is possible that some background or natural aerosols are already enough to allow EIE events to happen and that the aerosols from biomass burning/urban pollution are not actually making a difference. I don't want to get into more details since this would largely repeat many previous comments from myself and the other two reviewers. I do not intend to kill this manuscript since the dataset reported herein is really valuable. My suggestion is that the authors remove the arguments/conclusions on the causal relationships or physical links between EIE and biomass

burning/urban pollution aerosols. Otherwise, I still feel reluctant to support the publication of this manuscript.

*As the referee acknowledges, we went to great length to remove wording that would be mistaken as an attempt to unequivocally attribute not only EIE but all of the cirrus measured in this region. The revised manuscript was thoroughly screened for wording that would indicate attribution, which can only be done via modelling and not through an observational study.*

2) A minor issue: Line 625-627, "Of all the cells that had either clouds or CO anomalies, 52% had CO anomalies with no clouds, 10% had clouds with no CO anomalies, and 48% had concurrent observations of clouds and CO anomalies." The three percentages do not add up to 100%.

*Thank you, this has been corrected. The 48% should have been 38%.*

**Response to Referee #2**

1) The authors have captured the essence of my comments and those of other reviewers regarding the need to frame this study mechanistically before diving in to demonstrate relations to aerosols. Thus, the paper and its general focus are much better now. I understand the utility of the added case studies in addressing other reviewer concerns, even though I did not personally find them to add greatly to the paper. The introduction regarding nucleation processes still needs a little work in my opinion, and hence I make a few suggestions for consideration in laying out the basis for thinking about how the high ice concentrations could ensue.

*The authors would like to thank the reviewer for recognizing that we strove to address the comments and recommendations in the previous review. As acknowledged by this reviewer, the case studies were in response to another reviewer; however, we respectfully disagree and consider that their addition was important to show that our results and conclusions were not strictly based on a statistical association over the 9 years of measurements, but that statistics are founded on individual cases that can provide a direct connection between surface emission sources, particle composition, cloud microphysics and atmospheric dynamics. Without the detailed modeling that we mention in our comments to all reviewers (above), we believe that the statistical results supplemented by selected case studies do show compelling evidence that supports our conclusions.*

*The reviewer indicated making some suggestions for further consideration, but we are unable to determine from what is written below the specific suggestions referred to. Hence, in this new revision we have included additional text on homogeneous versus heterogeneous freezing and we have also strengthened the discussion of the importance of strong convection in the formation of these high ice concentration clouds.*

There is no discussion at all of homogeneous freezing, how it could come about, and whether composition really matters, preferring still to focus on heterogeneous nucleation. Instead, as in the response to reviews, the authors prefer to call them all ice-forming particles. That is not accurate. Some particles trigger ice formation, while ice formation ensues in others due to an ice formation process that cares very little about the particle except that there is dilute water in it.

*In the introduction we have expanded on the freezing types, and then again in the discussion we reiterate how homogeneous freezing may be equally likely as heterogeneous freezing under conditions of vigorous updrafts. We think that our use of the term "ice-forming particles" is more precise than the phrase that "particles trigger ice formation" or that some ice formation processes care little about the particle. Perhaps we are arguing semantics here but in the earth environment, with no particles on which to form, there will be neither water droplets nor ice crystals. Although in all cases of ice formation, the ice is forming on particles, we believe that our terminology of "ice forming particles" is a reasonable way to describe the particle on whose surfaces ice forms. In the revised version we have added a sentence to clarify this point.*

One reason that poor INPs such as biomass burning particles and urban pollution can impact ice formation in these clouds so strongly is most easily argued to be a process that is less deferential in selecting them for freezing (i.e., not due to the special properties of the particles). Excepting the regions influenced by mineral dusts, where one could plausibly say that thousands per liter might activate before the onset of homogeneous freezing as droplets are lofted, one apparently only needs high aerosol numbers and strong updrafts in order to promote more ice formation in the presence of more particles. This could be a working hypothesis for anyone pursuing this topic further, after reading this paper.

*We are in complete agreement with the reviewer that we expect, and would indeed welcome, modelling work motivated by this observational study to address some of the specific details that cannot be done with the datasets analyzed.*

For example, the argument now made is that sea salt can be ignored on a number basis alone. Well, there a second point made about its preferential liquid-phase scavenging in lower cloud regions, an argument that appears to ignore the role of cloud dynamics on impacting CCN activation and scavenging by any aerosol entering deep convection.

*We think that perhaps the reviewer misunderstood our point regarding the liquid-phase scavenging and upon reading that section again have edited it to make our point clearer, i.e., that marine aerosols, due to their size and hygroscopicity would be those that activate first and grow rapidly to raindrops that will remove them from the cloud before they are lofted to freezing temperatures.*

In general, the paper lacked an expressed appreciation for how dynamics can overcome restrictions on CCN activation. Under strong updrafts, one could easily posit that chemistry as a player in CCN hygroscopicity is likely irrelevant, and that is why BB and urban aerosols become important for anvil microphysics over the tropics. If this had been considered, Fan et al. (Science 359, 411–418, 2018) might have been referenced as a case for even small pollution particles as likely CCN that can freeze in upper cloud regions, due to cloud dynamics and the role of coalescence in driving supersaturations in elevated cloud regions. But I am not trying to rewrite the paper, only state what is apparent to a reader.

*We have added what we expect will satisfy the reviewer's interest in making the role of strong updrafts more obvious. Our use of the maps of upper air divergence and the other analysis tools that we employed to underscore this role, indicate our awareness of the role of strong updrafts.*

Line 21: Acronym SOFT-IO not defined in abstract.

*The paper by Sauvage et al., 2017 introduces the name "SOFT-IO version 1.0" to describe the software developed which combines the FLEXPART (Stohl et al., 2005) Lagrangian dispersion model with the inventory of the Emissions of atmospheric Compounds & Compilation of Ancillary Data (ECCAD) emission database (Granier et al., 2012). It does not indicate that SOFT-IO is an acronym, but rather the way to identify the software developed in their study, which is now available to users.*

Lines 30-31: Heterogeneous and homogeneous freezing. Saying both would make it clearer that both are likely involved.

*We have added "Heterogeneous and homogeneous" to the text.*

Lines 81 paragraph: It is a little odd to start this paragraph this way, since you have already mentioned heterogeneous and homogeneous freezing above this point. It would be more appropriate to speak to both mechanisms then at this point. The paragraph only mentions heterogeneous INP concentrations and ice crystal concentrations, ending presently with a very nice statement about that. But why not say that homogeneous freezing would be driven by strong updrafts that send condensed water not already frozen by limited heterogeneous freezing or limited consumption by ice growth into the regime where remaining drops can freeze? Noting the role of updraft on activating particles pre- and post-coalescence could add some context as to why this mechanism might be particularly powerful in affecting upper anvil ice concentrations.

*We believe that perhaps the reviewer didn't understand the purpose of this paragraph since in the discussion of the previous paragraph, we were only referring to homogeneous freezing whereas in the paragraph beginning on line 81, we are talking about depositional nucleation. Yes, when we reference Krämer et al. (2016) and cirrus of in situ origin, we*

*know they are talking about depositional nucleation but this paragraph further explains why that is unlikely to be the mechanism forming the ice in the tropical clouds. That being said, we realize that we have not made this very clear and have added a couple of sentences to link this type of ice formation to the in situ type of cirrus formation.*

*In addition, we have added to the previous paragraph words to the same effect as suggested by the reviewer, reinforcing the important role of strong updrafts to loft liquid water that has not already frozen heterogeneously into regions where it freezes homogeneously. That is a point that we failed to include in our discussion, i.e., that the two mechanisms are not mutually exclusive.*

Lines 105-107: Not a nuanced discussion. Would one not expect some major fraction of all particles to be available as CCN from combustion sources, but some more limited fraction available as INPs prior to onset of homogeneous freezing conditions? I say this only because it otherwise sounds like very little is known, but that is not the case.

*We agree that this single sentence by itself offered little information. We have added a bit more context as suggested by the reviewer, in particular here is where we highlight that it might not be the composition of the combustion particles as much as their sheer number that get activated due to high supersaturations generated in strong updrafts.*

Line 125-127: Contrast the above discussion with the apparent need to state that rBC particles are coated. This is probably a nearly irrelevant factor since supersaturations could be high in deep convective updrafts (especially following initial coalescence). Also, is it not the case that rBC is but a small fraction of all biomass burning particles, all of which are available to act as CCN? You later rule out sea salt for similar number-based arguments.

*We agree and here we have reiterated what we have now added in the previous discussion on the role of combustion particles that it is likely the number of BC and not their size or composition that could contribute to high ice in the presence of strong updrafts.*

Line 230: "Concentrations" or "Mixing ratios"?

*We have changed concentrations to mixing ratios here and throughout when talking about CO in order to not confuse the reader with number concentration.*

Lines 285-292: The statements made on not considering sea salt remains speculative. It will "likely" be less of a factor would solve this for now.

*Changed to "..it will likely be less of a factor.."*

4) Results

Line 317: 0.01 cm-3 as a lower bound on ice concentrations. I thought the lower limit of detection used in this study was 50 per liter?

*We have changed 0.01 to 0.05.*

Lines 414-415: Dust acting as good INPs by "deposition" of water vapor to their surfaces? If all of the cirrus discussed were of liquid-origin (also stated as a conclusion on lines 584-585), then why mention a mechanism that is unrelated to liquid droplet activation?.

*We have now included words to the effect that dust particles, although very good INP will also activate as CCN under the high updraft scenario.*

Lines 440-441: This repeats the same point about dust, but is better-spoken here. Could say this just once, in one spot or the other.

*We have added here a sentence that refers back to what we added after line 415.*

Line 454: Second mention of sulfate "mixed with dust", but it is unclear why this is at all important, or if it is important (which I doubt, if CCN activation is the concern –i.e., big dust > smaller CCN for activation under strong forcing, all else being equal and no matter the sulfate).

*We agree and have now reworded this statement with a reiteration that even though these mixture make them better CCN, under vigorous updrafts they would activate regardless of their composition.*

Line 577: Discussion of ice crystal residuals as "ice nuclei". You could stop that phrase at their "composition", since "ice nuclei" infers a component that freezes heterogeneously. It could say ice crystal nuclei, but ice residual nuclei is already stated, and a better way to discuss it.

*Agreed. We have removed "that could identify the composition of the ice nuclei"*

Lines 587-588: "…cloud chamber and field studies have shown that some fraction of the BB and UP aerosol are hygroscopic and can serve as CCN." Just how hygroscopic do they really need to be in these circumstances? Would kappa of 0.1 not be sufficient? See Twohy et al. (2021; https://doi.org/10.1029/2021GL094224) on how easily smokes are activated even in modest cumuli.

*Now that we have clarified in the introduction and throughout the text the importance of the high updraft, we have inserted again that the composition might not matter under those conditions.*

Line 664: Indeed, any boundary layer aerosols are lofted by strong convergence, and hence, one expects pollution, smoke, dust, and even sea spray particles (primary and secondary formed ones) to influence convective cirrus that dominate in the tropics. This paper confirms that.

*Agreed.*

---

## Author Response (AR3)

Once again we thank the reviewer for taking the time to read through our revised manuscript and would like to respond to the final comments that were made to our revision.

In particular, the reviewer says the following: As for the assertion to another reviewer that all attribution of EIE to certain types of aerosols has been carefully removed, I can only point to line 696-697 of the manuscript that states "The current study has established a pathway that associates extreme ice concentrations with the surface emissions of particles from biomass burning and fossil fuel combustion." There it is, for all to potentially mis-reference. While this admittedly follows a paragraph that emphasizes the association with convective dynamics, the list of subsequent remaining questions does not mention the critical role of deep convection, nor leave a question as to whether the aerosol sources matter in general for the production of EIE.

We respectfully disagree with the reviewer's concern that our conclusion regarding the association between extreme ice concentrations and surface aerosol emissions will be mis-referenced. Nevertheless, following the reviewer's recommendation, we have now added a final bullet to the unresolved questions: "Does deep convection with strong updrafts minimize the importance of aerosol composition?"

As to the reviewer's second statement: I remain completely baffled by authors' understanding and discussion of homogeneous freezing as happening on particle "surfaces" in their response. A surface does not exist for a dissolved particle. If there is anywhere in aerosol-cloud interactions research that careful terminology is needed, it is in the area of distinguishing heterogeneous and homogeneous nucleation processes and the specific role or not of particle surfaces in the formation of cirrus clouds. It if were stated this way in the paper, it would set the ice nucleation community back years in their efforts to use concise language, I would say. Nevertheless, the point is moot, because I could find nowhere in the paper where such a discussion now occurs, and what is clarified now is quite sufficient for distinguishing processes that relate to aerosol properties, and evolve and impact cirrus, in very different ways.

It appears that no further action in the manuscript is needed.